# Hallmarks of primary neurulation are conserved in the zebrafish forebrain

Jonathan M. Werner[1,2], Maraki Y. Negesse [1,2], Dominique L. Brooks[1], Allyson R. Caldwell[1], Jafira M. Johnson[1] & Rachel M. Brewster [1✉]

Primary neurulation is the process by which the neural tube, the central nervous system precursor, is formed from the neural plate. Incomplete neural tube closure occurs frequently, yet underlying causes remain poorly understood. Developmental studies in amniotes and amphibians have identified hingepoint and neural fold formation as key morphogenetic events and hallmarks of primary neurulation, the disruption of which causes neural tube defects. In contrast, the mode of neurulation in teleosts has remained highly debated. Teleosts are thought to have evolved a unique mode of neurulation, whereby the neural plate infolds in absence of hingepoints and neural folds, at least in the hindbrain/trunk where it has been studied. Using high-resolution imaging and time-lapse microscopy, we show here the presence of these morphological landmarks in the zebrafish anterior neural plate. These results reveal similarities between neurulation in teleosts and other vertebrates and hence the suitability of zebrafish to understand human neurulation.

[1] Department of Biological Sciences, University of Maryland Baltimore County, Baltimore, MD 21250, USA. [2] These authors contributed equally: Jonathan M. Werner, Maraki Y. Negesse. ✉email: brewster@umbc.edu

Primary neurulation is the process by which the neural tube, the precursor of the brain and spinal cord, is shaped from the neural plate. These morphogenetic events have mostly been studied in amniotes (mouse and chick) and amphibians (frogs), where conserved mechanisms were identified. Following neural induction, the neural plate (Fig. 1a) narrows and elongates by convergent extension movements[1,2]. The morphology of the neural plate is further changed in a biphasic manner[3]. The first morphological event is the formation of the medial hingepoint (MHP), which bends the flat neural plate into a V shape, forming the neural groove (Fig. 1b). Neural Folds (NFs) at the edges of the neural plate subsequently elevate, a process that is further driven by head mesenchyme expansion[4]. In the second phase, the neural plate folds at paired dorso-lateral hingepoints (DLHPs), bringing the NFs closer together (Fig. 1c). The NFs eventually meet and fuse at the midline, completing neural tube formation. The dorsal midline is subsequently remodeled to separate the inner neuroectoderm from the outer non-neural ectoderm fated to become the epidermis (Fig. 1d). In the cranial region of amniotes, neural tube closure is initiated at several sites and extends via a zippering process from these closure points to seal the neural tube.

Incomplete cranial neural tube closure occurs frequently, resulting in anencephaly and exencephaly[3].

Several cellular mechanisms have been identified that contribute to the bending of the neural plate, among which apical constriction is the most well studied[5–8]. F-actin accumulates at the apical cortex of neuroectodermal cells and contracts, thereby reducing the cell apex (Figure 1b1). The contraction is driven by myosin that colocalizes with F-actin at the apical pole. Disruption of F-actin using drug inhibitors or genetic tools causes severe cranial neural tube defects[5,9–14]. Similarly, treatment with blebbistatin, an inhibitor of non-muscle myosin II (NMII) activity, impairs apical constriction in the superficial layer of the *Xenopus* neural plate[15].

In contrast to hingepoint formation, the cell-intrinsic mechanisms that shape NF cells are less understood. The NFs are bilaminar, consisting of a layer of neuroectoderm capped by a layer of non-neural ectoderm[16], a topology that is acquired via epithelial ridging, kinking, delamination, and apposition[16,17]. NF fusion involves the formation of cellular protrusions that span the midline gap and establish the first contact with NF cells from the contralateral side[18–20] (Figure 1c1). There is little consensus on the cell type (neuroectoderm or non-neural ectoderm) that

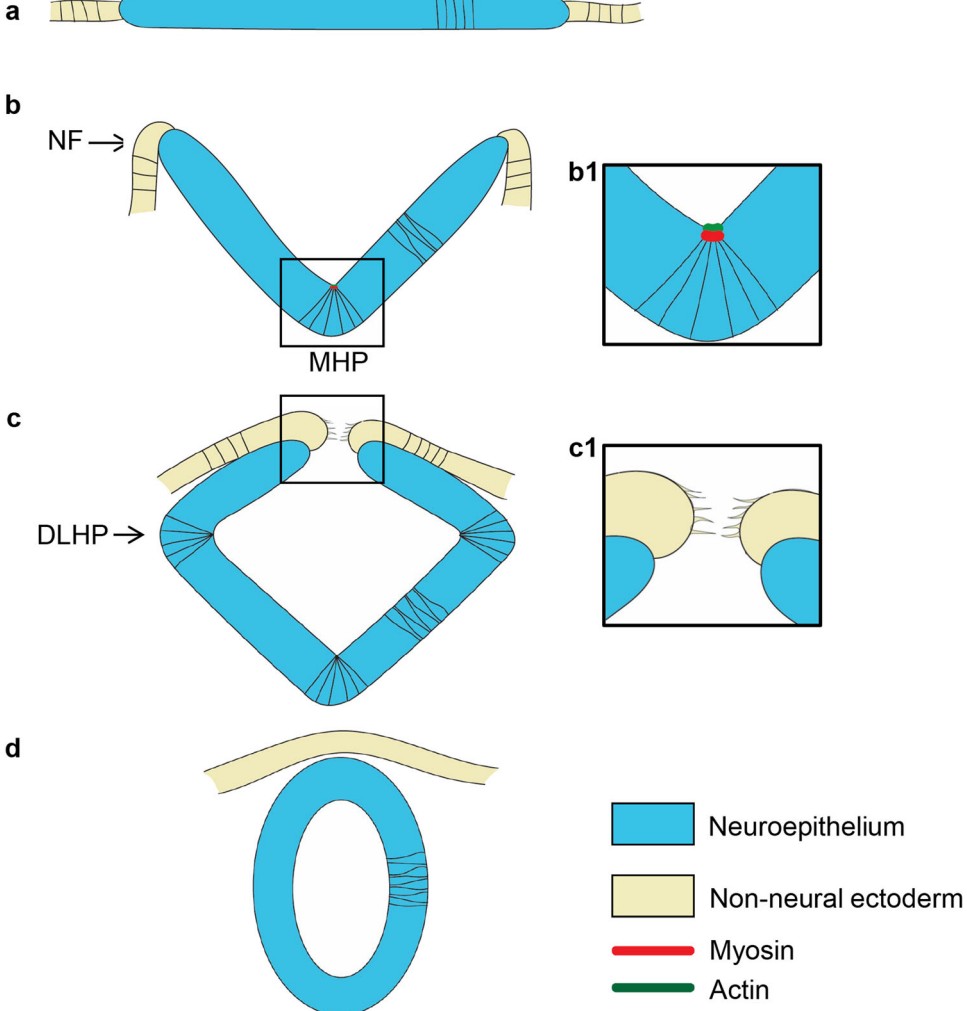

**Fig. 1 Neurulation in amniotes. Cross-sectional illustration of stages of neurulation in amniotes. a** The neural plate and adjacent non-neural ectoderm. **b** Medial hingepoint formation shapes the neural groove and elevates the neural folds. **b1** Illustration of medial hingepoint cells that are apically constricted and enriched for actomyosin at their apex. **c** Dorso-lateral hingepoint formation brings the neural folds in close apposition. **c1** Filopodial extensions establish contact between neural fold cells across the midline. In the mouse forebrain, the first contact is established between neuroectodermal cells. **d** The neural folds fuse medially, separating the epidermis from the neural tube. *DLHP* dorso-lateral hingepoint, *MHP* medial hingepoint, *NF* neural fold.

generates the cellular protrusions as it varies depending on the species and the axial level. In the mouse forebrain, the initial contact is made by neuroectodermal cells[18,21–24]. Disruption of F-actin blocks the formation of protrusions and prevents NF fusion[20], revealing the central role of these cellular processes.

Although more anterior regions of the neural tube undergo primary neurulation as described above, most vertebrates also exhibit a distinct type of neural tube formation in more posterior regions, termed secondary neurulation. During this process, a mesenchymal cell population condenses into a solid neural rod that subsequently epithelializes and forms a central lumen[25,26]. In *Danio rerio* (zebrafish), the neural tube in the hindbrain and trunk region initially forms a solid rod that later develops a lumen, a process seemingly analogous to secondary neurulation[27]. However, examination of the tissue architecture in zebrafish[28–30] and other teleosts[31,32] revealed that the neural rod is shaped by infolding of a neural plate (albeit incompletely epithelialized), which best fits the description of primary neurulation[33]. Despite this evidence, differences in tissue architecture, the multi-layered organization of the neural plate and the apparent lack of hingepoints, neural groove and NFs are difficult to reconcile with a mode of primary neurulation and have contributed to the persistent view that neural tube formation in teleosts is different than in other vertebrates[34–36].

We show here that, in contrast to the zebrafish hindbrain and trunk region, the process of neural tube formation in the forebrain exhibits some of the hallmarks of primary neurulation. We observe the presence of hingepoints and NFs in the epithelialized anterior neural plate (ANP), demonstrate that formation of the MHP involves oscillatory apical contractions that progressively reduce the apical surface, and show that disruption of myosin function impairs apical constriction of these cells and NF convergence. We further show that neural tube closure is initiated at two separate sites in the forebrain and that fusion of the NFs is mediated by filopodial-like extensions of neuroectodermal cells that bridge the midline. These findings identify conserved mechanisms of primary neurulation that were previously overlooked in teleosts and support the suitability of zebrafish for understanding the etiology of human neural tube defects.

## Results

### Precocious epithelialization of the ANP is associated with bending.

The zebrafish ANP is quite distinct from the neural plate in more posterior regions, as it undergoes precocious epithelialization[37]. To assess whether epithelialization correlates with a change in the mode of neurulation, we examined the morphology of the neuroectoderm in optical cross-sections at developmental stages 2–10 somites (som). We observed that at 2–5 som the ANP has a V shape marked by a medial neural groove (black arrowhead) flanked by the elevated lateral edges of the ANP (white arrowhead), which are reminiscent of NFs (Fig. 2a, b). By 7 som, the groove is no longer visible and the elevated edges of the ANP have fused medially (Fig. 2c, white arrowhead) and form a dorsal bulge (Fig. 2c, d, white arrowhead). These observations suggest that the ANP bends at the midline, which correlates with elevation of the lateral edges of the ANP.

### The ANP is multi-layered and gives rise to the eyes and forebrain.

The central part of the ANP is fated to become the eyes, whereas its lateral edges produce the dorsally-located telencephalon and the ventral-most ANP region gives rise to the hypothalamus[37,38]. To gain a better understanding of the morphogenetic events that shape the ANP, we examined transverse views of transgenic Tg[*emx3:YFP*] embryos at developmental stages 2, 5, 7, and 10 som, in which telencephalon precursors are

labeled[39]. We observed, as previously described[37], that the ANP is a multi-layered tissue, with a mesenchymal superficial core and a marginal or deep layer. These layers will henceforth be referred to as the superficial and deep layers, respectively (s and d in Fig. 2e, f). The YFP-positive lateral edges of the ANP elevate, migrate over the eye field and fuse medially at 7 som to form the telencephalon (Fig. 2g, h). By 7 som the multi-layered ANP resolves into a single-cell layered neuroectoderm (Fig. 2g). Although morphological changes and cellular dynamics that form the eyes are quite well understood[37,38], the accompanying events that shape the forebrain are for the most part unknown and the focus of the current study.

### Presence of hingepoints and NFs in the ANP.

To investigate whether the ANP bends and folds around hingepoints to bring NFs in close apposition, we examined the cytoarchitecture of this tissue in embryos at stages 2–10 som, labeled with phalloidin (filamentous (F) actin), anti-Sox3 (neural cells), and anti-p63 (epidermal cells, nuclear label) (Fig. 2i–l).

We found that at 2 som, neural cells appear mesenchymal with no visible polarized enrichment of F-actin (Fig. 2i), consistent with previous observations[37]. The epidermis at this stage is in a far lateral position (white arrowhead in Fig. 2i).

Between 2 and 5 som, clusters of cells in the superficial and deep layers of the eye field have been reported to undergo early epithelialization[37], which we confirm here with foci of F-actin enrichment in the medial/superficial region (M in Fig. 2j, m) and in two dorso-lateral clusters in the deep marginal layer (DL in Fig. 2j, m). The apical surfaces of the superficial cells appear to constrict and orient towards the midline, resulting in the formation of a medial neural groove (asterisk in Fig. 2m). Similarly, the paired dorso-lateral clusters of cells in the deep layer are also apically constricted and the neuroectoderm folds sharply at this level (red arrows in Fig. 2m). These data show that the medial and dorso-lateral cells are enriched for apical F-actin and suggest that they function as hingepoints. Similar to amphibians whose neural plate is bilayered[40], the putative MHP in the zebrafish ANP forms in the superficial layer and is therefore more dorsally positioned than in amniotes.

The lateral edges of the ANP are bilaminar at 5 som, consisting of a layer of neuroectoderm cells capped by a layer of p63-positive non-neural ectoderm (and Sox3/p63-negative olfactory placodal cells bridging the two layers), indicative of a NF structure (Fig. 2j, m). The YFP-positive ANP cells in Tg[*emx3:YFP*] embryos correspond to the neuroectoderm component of the NF (Fig. 2f), revealing that the tip of the NF gives rise to the telencephalon. At 4 som, the YFP-positive region of Tg[*emx3:YFP*] embryos extends the length of the forebrain (Fig. 2o), delineating the anterior-posterior range of the NFs.

The NF and putative hingepoints are transient as they are no longer observed in 7 som embryos. By this stage, the tips of the NFs have converged medially and fused, forming the telencephalon (Fig. 2k). These cells are enriched in apical F-actin at 10 som, suggesting that they epithelialize (Fig. 2l). The non-neural ectoderm still occupies a lateral position at 7 som (Fig. 2k, arrowheads), however by 10 som these cells migrate and fuse dorsally (single arrowhead in Fig. 2l), indicating that, as observed in mice, the neuroectodermal component of the NFs meet first (Fig. 1)[21,22]. Measurements of the distance between the medial-most p63-positive domain and the dorsal midline at stages 2–7 som indicate that the non-neural ectoderm portion of the NFs converges steadily towards the midline and may provide a lateral force that contributes to the displacement of NFs (Fig. 2n).

These observations reveal that transient medial and dorso-lateral epithelialized cell clusters may be the functional

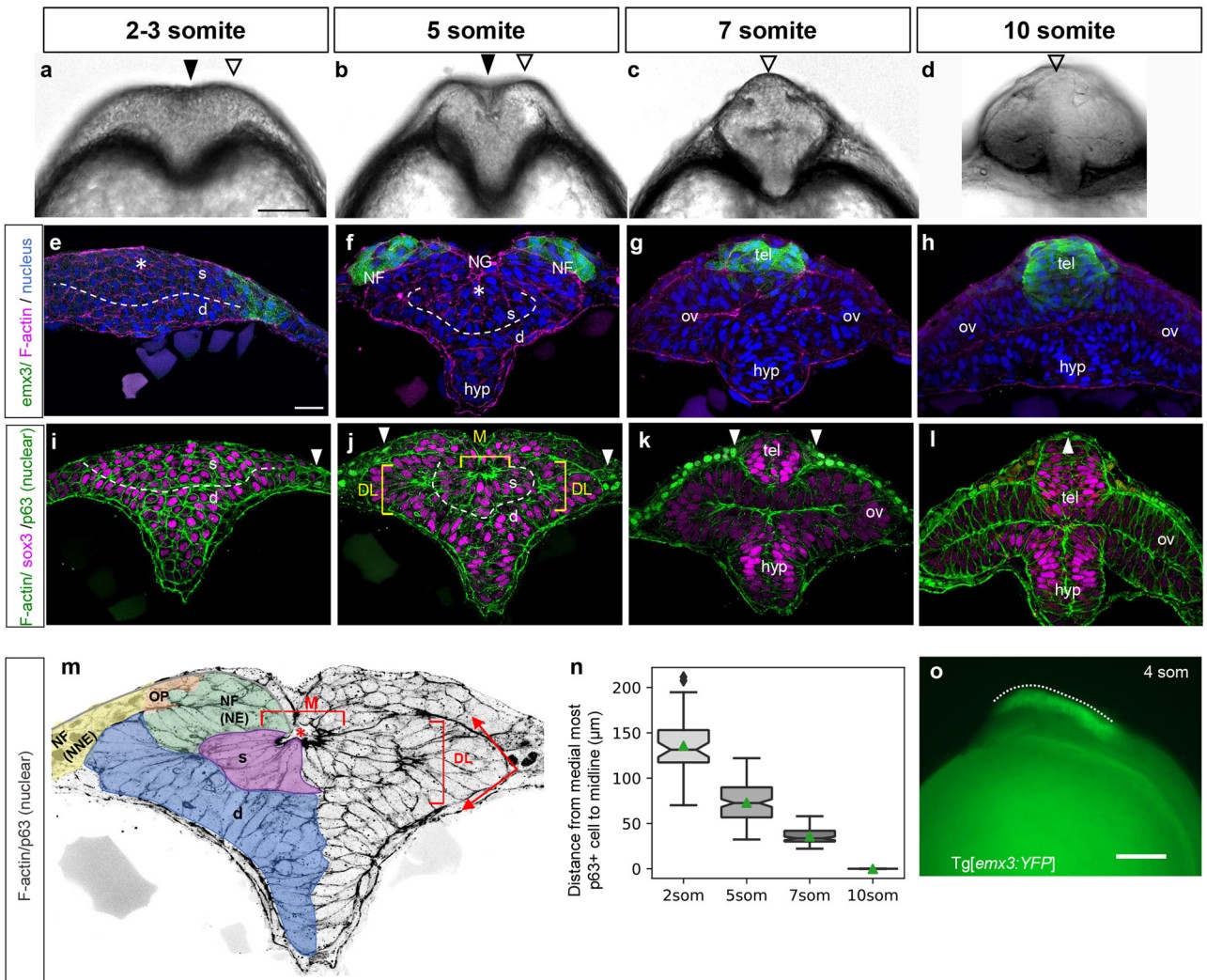

**Fig. 2 Hingepoints and neural folds contribute to forebrain morphogenesis. a–d** Optical sections at the level of the forebrain of WT embryos at the 2–3 som (**a**), 5 som (**b**), 7 som (**c**), and 10 som (**d**) stages. **e–l** Transverse sections through the ANP of 2–3 som (**e, i**), 5 som (**f, j**), 7 som (**g, k**) and 10 som (**h, l**) embryos. **e–h** Tg[*emx3:YFP*] embryos labeled with anti-GFP (green), phalloidin (F-actin, magenta), and DAPI (nuclei, blue). **i–l** WT embryos labeled with phalloidin (F-actin, green), anti-Sox3 (magenta), and anti-p63 (nuclear label, green). **m** Higher magnification image of **j**, gray scaled to reveal F-actin and p63 and pseudo-colored - color code: light blue: superficial eye field cells, dark blue: deep layer eye field cells that apically constrict to form the optic vesicles, green: neural component of the neural fold, orange: olfactory placode (Sox3/p63-negative cells), yellow: non-neural component of the neural fold. **n** Measurements of neural fold convergence, scored as the distance between the medial-most p63-positive cells on either side of the midline at different developmental stages. Notches depict the 95% confidence interval around the median and the green triangle depicts the distribution mean. 2 som: 48 measurements from 14 embryos, mean = 136; 5 som: 144 measurements from 16 embryos, mean = 73.0; 7 som: 87 measurements from 10 embryos, mean = 36.0; 10 som: p63 domain is fused, no measurements. Statistical analysis: Mann–Whitney *U* tests, two-sided; 2 som vs 5 som: P = 1.18e$^{-21}$, AUC = 0.961; 2 som vs 7 som: *P* = 8.35e$^{-22}$, AUC = 1.00; 5 som vs 7 som: *P* = 2.14e$^{-31}$, AUC = 0.958. **o** Side view of a 4 som Tg[*emx3:YFP*] embryo. *s* superficial layer, *d* deep layer, *NF* neural fold, *NG* neural groove, *hyp* hypothalamus, *tel* telencephalon, *ov* optic vesicle, *DL* dorso-lateral hingepoints; *M* medial hingepoint, *NE* neural ectoderm, *NG* neural groove, *NNE* non-neural ectoderm, *OP* olfactory placode. Annotations: black arrowhead = median groove, white open arrowhead = elevated neural fold-like structure, dashed line = separation of the deep and superficial layers; brackets = hingepoints; dotted line = A-P range of the neural folds; white arrowhead = medial-most epidermis; red asterisk = neural groove. Scale bars: **a** and **o** = 100 µm, **e** = 25 µm.

equivalents of the MHP and paired DLHPs of amniotes (and will be referred to henceforth as such), as they form at the right time and place to contribute to the formation of the neural groove (MHP), bending of the neuroectoderm and medial convergence of the NFs (DLHPS).

**Cell shape changes underlying dorso-lateral hingepoint and neural fold formation.** To image cells at higher resolution, we mosaically expressed membrane-targeted GFP (mGFP) and

imaged embryos at the 2–10 som stages in transverse sections (Fig. 3). 3D reconstructions of some of these images were generated to visualize the spatial relation between labeled cells (Supplemental Movie 1, 2).

At 2 som, cells in the deep layer have a columnar shape with one end in contact with the basal lamina and a future apical surface oriented towards the midline (Fig. 3a). These cells extend membrane protrusions into the superficial layer, which may promote radial intercalation (inset in Fig. 3a).

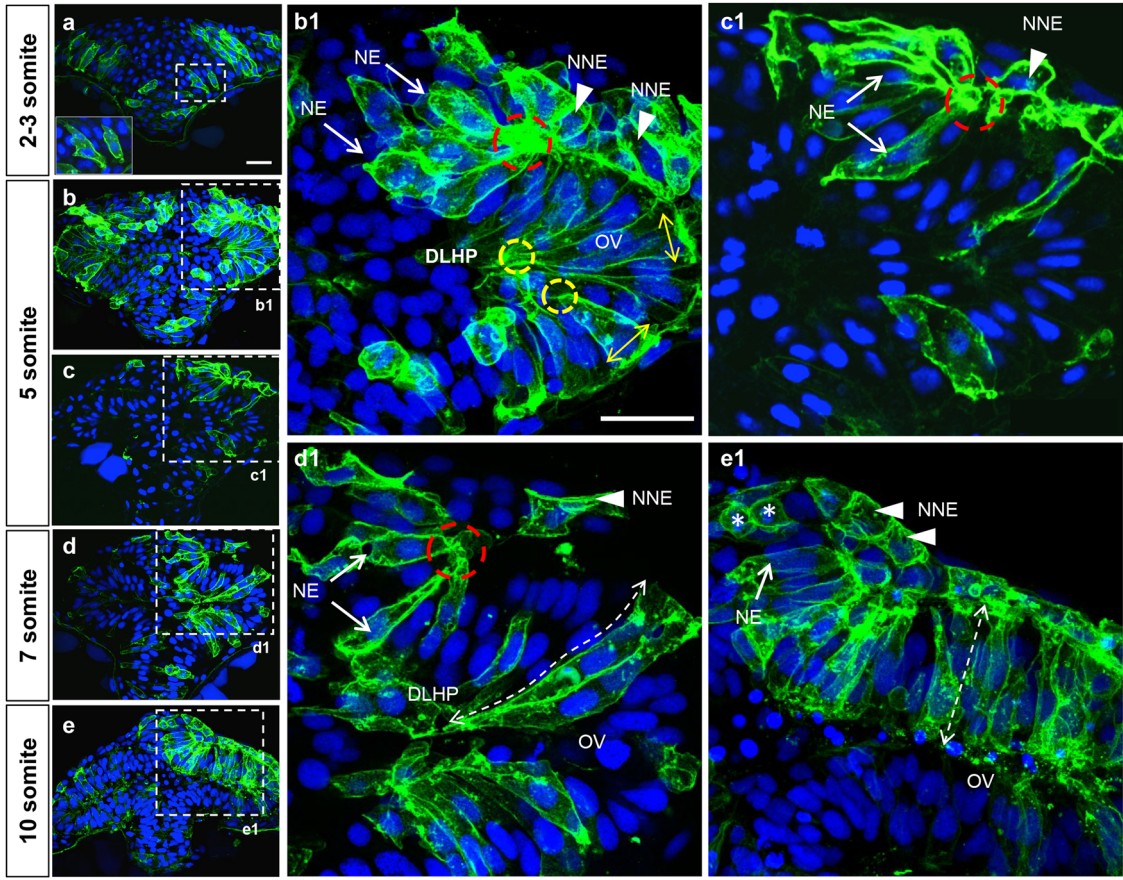

**Fig. 3 Cell shape changes in the deep layer of the ANP that contribute to DLHP and neural fold formation. a–e** Transverse section, at the level of the forebrain, of embryos at the 2–3 (**a**), 5 (**b**, **b1**, **c**, **c1**), 7 (**d**, **d1**), and 10 (**e**, **e1**) somite stages mosaically expressing mGFP (green) and labeled with the nuclear marker DAPI (blue). The inset in **a** is a higher magnification of dashed area in **a**. **b1–e1** Higher magnifications of regions delineated by dotted lines in **b–e**. Annotations: red dashed circle = basal constriction of NE component of neural fold; yellow circle = apical constriction of DLHP cell; arrows = NE component of neural fold; arrowheads = NNE component of neural folds; double dashed arrow = elongated deep cells of the optic vesicle, asterisks = dividing cells in the prospective telencephalon. Scale bars: 25 μm in **a** and **b1**.

At 5 som, cells in the dorso-lateral deep layer have undergone apical constriction and basal expansion, forming DLHPs (yellow dotted circles and double arrowheads in Figs. 3b1; Supplemental Movie 1). These cell shape changes may initiate the outpocketing of the optic vesicles (ov in Figs. 3b, b1). The bilaminar organization of the NFs is clearly visible at this stage. Neuroectodermal cells within the NFs (arrows in Figure 3b1 and c1) are elongated and their basal poles are constricted, giving them the appearance of fanning out from a focal point (red circles in Figure 3b1 and c1). Furthermore, their plasma membrane is ruffled, indicative of protrusive activity (Supplemental Movie 1).

By 7 som (Fig. 3d, Supplemental Movie 2), neuroectodermal cells of the NFs have reached the dorsal midline. They maintain their organization with basal poles constricted and clustered at a focal point on the basement membrane (red circle in Figure 3d1). 3D reconstructions reveal that these cells are finger-shaped as they extend across the dorsal midline. DLHP cells maintain their apical constriction/basal expansion and further elongate, contributing to the expansion of the optic vesicles (dotted double arrowhead in Figure 3d1).

At 10 som (Figs. 3e, e1) deep cells in the eye field and NFs have a columnar, epithelial organization. Consistent with previous findings, eye field cells shorten along their apico-basal axis by contracting their apical processes and coincidently transition to a more dorso-ventral orientation[37] (dotted double arrow in Figure 3e1). Cuboidal non-neural ectoderm cells cover the dorsal surface of the newly formed telencephalon (arrowheads in Figure 3e1).

The cell shape changes in the deep layer of the neuroectoderm were quantified by measuring cell length and the apico:basal ratio of mGFP-labeled cells located at different positions along the medio-lateral axis of the ANP at 2, 5, and 7 som (Fig. 4). These data reveal that the average length of DLHP cells increases while their apical:basal ratio decreases between 2 and 7 som, coincident with optic vesicle evagination. They also confirm that neuroecto-dermal cells of the NFs adopt the opposite configuration.

To capture the dynamics of NF formation, we performed transverse view time-lapse imaging of WT embryos expressing membrane-targeted Kaede (mKaede) or Tg[emx3:YFP] transgenic embryos expressing membrane-targeted RFP (mRFP) (n = 4 embryos, Fig. 5; Supplemental Movies 3 and 4). We observed that as the NFs elevate (blue dotted line in Fig. 5a–c), the basal surface of NF cells narrows (red outline and arrowhead in Figure 5a1, c1). Interestingly, this cell shape change appears restricted to cells within the emx3 expression domain (Figs. 5d–f1).

These findings indicate that zebrafish DLHP cells adopt a wedge shape similar to their counterparts in amphibians and amniotes, which may facilitate epithelial folding. They further reveal that NFs have a "reverse hingepoint" cytoarchitecture, with a narrow basal pole anchored on a focal point and an expanded apical surface.

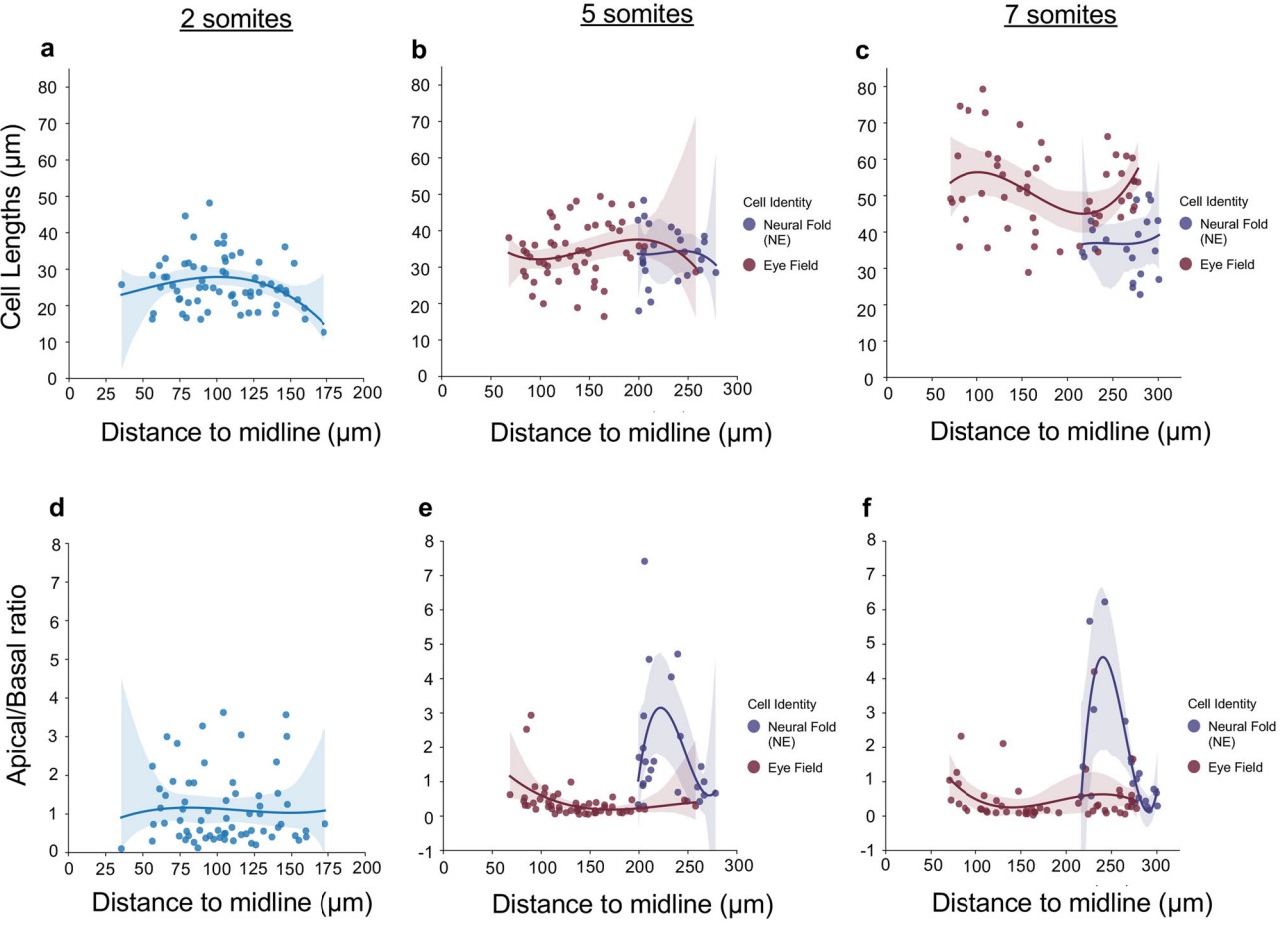

**Fig. 4 Measurements of cell shape changes in the deep neuroectodermal layer and neural folds. a–c** Measurements of cell lengths (μM, *Y* axis) of mGFP-labeled cells at different positions relative to the ANP midline (μM, *X* axis). Measurements begin in the region fated to become the optic vesicles. **d–f** Measurements of the apico:basal surface ratio of mGFP-labeled cells at different positions relative to the ANP midline (μM, *X* axis). Measurements begin in the region fated to form the optic vesicles. The original scatter plot was fitted to a 3rd degree polynomial. Confidence intervals for the polynomial are 95% and were calculated with bootstrap sampling (*n* = 1000). 2 somites: *n* = 66 cells from three embryos, 5 somites: 77 cells from 3 embryos, 7 somites: 72 cells from 3 embryos. Color code: light blue = neuroectodermal cells of the deep layer of 2 som stage embryos that are not yet identifiable based on cellular morphology; red = cells that form the optic vesicles; purple = cells that form the neuroectodermal component of the neural folds.

**Fate mapping of MHP cells**. The MHP is a transient structure, given that it is no longer observed by 7 som. To fate map this cell population, we photoconverted mKaede in the dorsal and medial region of the ANP (*n* = 2 embryos, Supplemental Movie 3 and Fig. 5g–i). We observed that these cells sink inwards and become incorporated into the deep layer of the eye field. These findings are consistent with a previous study showing that superficial cells radially intercalate between deep marginal cells[37]. Zebrafish DLHPs do not undergo these cellular rearrangements as they form in the deep layer.

**MHP cells constrict apically and elongate**. Although mosaic expression of mGFP enabled high-resolution imaging of DLHPs and NFs, it resulted in fewer cells labeled in the medial superficial layer, where the MHP forms. Phalloidin labeling was therefore used to image these cells in 2 and 5 som embryos.

At the 2 som stage, some medial/superficial cells immediately below the enveloping layer (EVL) are apically constricted, forming the MHP (Figs. 6a–a2). By 5 som, MHP cells appear more densely packed, the majority of them are apically constricted and oriented towards the midline. Concomitant with these cell shape changes and medial convergence of NFs, MHP cells shift to a more ventral position (Figs. 6b–b2). A small

opening, the neural groove, is observed immediately above the MHP at this stage (NG in Figure 6b1).

Measurements of the apico:basal surface ratio of MHP cells in 5 som embryos confirmed that they are wedge-shaped (Figure 6c1). Similar to amphibians[41], apical constriction appears tightly coupled to cell elongation, as the length-to-width ratio (LWR) of MHP cells increases significantly between 2 and 5 som (Figure 6 c2). Thus, both apical constriction and cell elongation appear to shape the MHP and contribute to tissue-level morphogenesis.

Another contributing factor to hingepoint formation in amniotes is the basal location of nuclei in both the medial and lateral hingepoints[42,43]. However, in zebrafish embryos, the relative position of the nucleus in medial superficial cells at 2 and 5 som does not change significantly and is therefore unlikely to contribute to wedging of these cells.

**Oscillatory constrictions reduce the apical surface of MHP cells**. To gain a better understanding of the dynamics of apical constriction and cell internalization, the ANP of embryos ubiquitously expressing mGFP was imaged from a dorsal view using time-lapse microscopy between 2 som and 4 som (*n* = 2 embryos, Fig. 6d and Supplemental Movie 5). The focal plane was set immediately below the EVL, at the level of the MHP. These

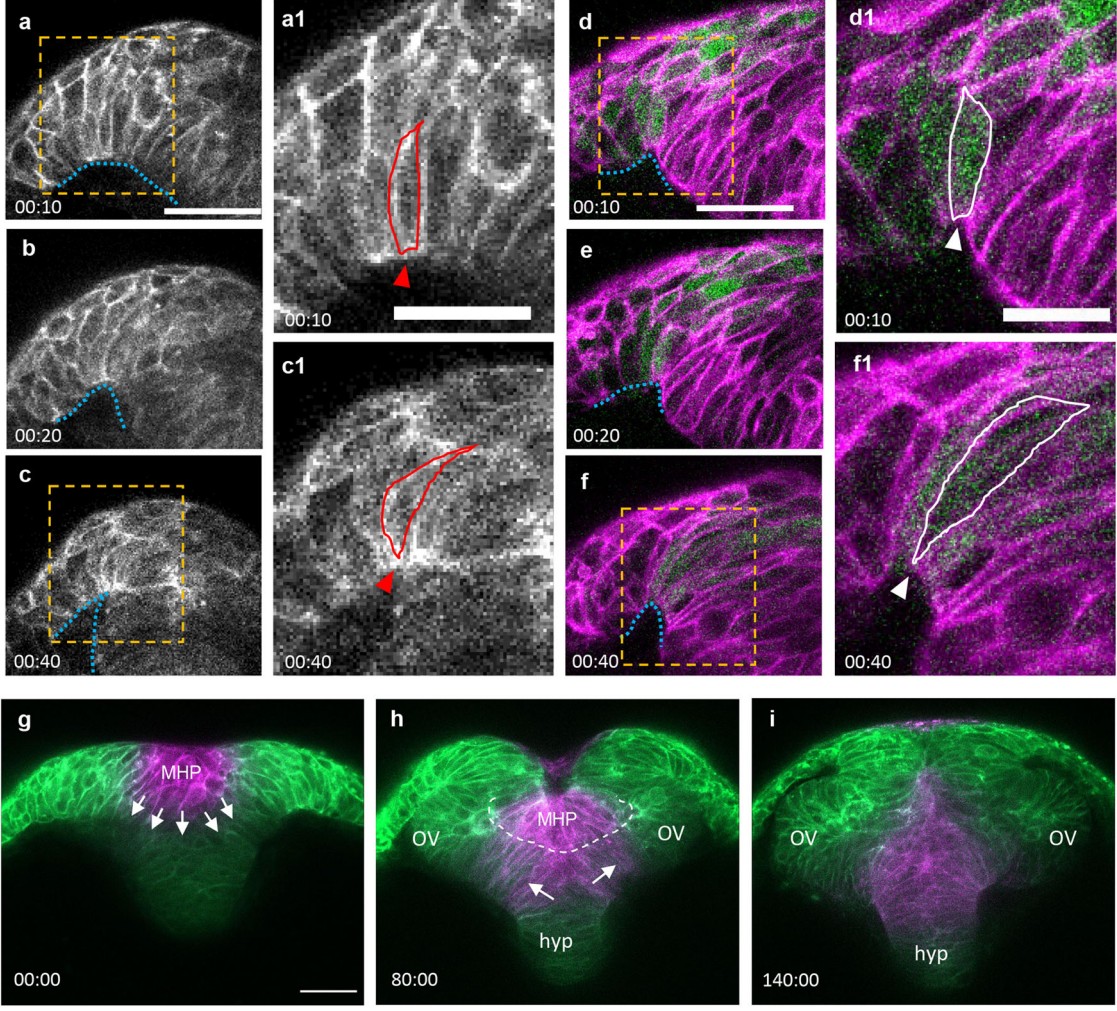

**Fig. 5 Dynamics of neural fold formation and MHP intercalation. a–f** Time-lapse movie frames of the ANP, from a transverse view. **a–c** Still frames of an embryo expressing membrane Kaede (mKaede). **d–f** Still frames of a Tg[*emx3:YFP*] embryo expressing membrane RFP (mRFP, pseudo labeled magenta) and YFP (green). Yellow boxes in **a**, **c**, **d**, and **f** identify magnified areas in **a1**, **c1**, **d1**, and **f1**. **g–i** Still frames of an embryo expressing mKaede (green), in which the MHP cells have been photoconverted (magenta) to follow their fate. *MHP* medial hingepoint, *OV* optic vesicles, *hyp* hypothalamus. Annotations: blue dotted line: outlines the basal side of the neural folds; red and white lines identify individual neural fold cells; arrowhead: indicate narrowing surface in neural folds cells; arrows: show direction of intercalation of MHP cells into the eye field; dotted white line: separation of the superficial and deep layers. Scale bars: 50 μm in **a**, **d** and **g**; 25 μm in **a1** and **d1**.

movies revealed clusters of medially located cells that undergo progressive cell surface reduction (color-coded in Figure 6d1–d7), whereas the surface area of more lateral cells remained unchanged for the duration of imaging (yellow asterisks in Figure 6d1–d6). The EVL came into focus immediately posterior to the cells with reducing apices. As the EVL is drawn inward as a result of its close contact with apically constricting MHP cells (Figs. 6b–b2), we surmise that apical constriction proceeds in a posterior-to-anterior direction. In later movie frames, EVL cells are no longer observed within the field of view, coinciding with the proximity of the NFs to the midline (Supplemental Movie 5). This indicates that medial EVL cells eventually lose contact with the MHP and return to their original position, coinciding with NFs fusion (Fig. 2j, k).

To further examine the dynamics of apical constriction, the surface area of superficial ANP cells was measured at shorter intervals. Midline (MHP) cells (20 μm on either side of the midline, Fig. 7a, e) and cells adjacent to the midline (>20 μm on either side of the midline, Fig. 7b) were scored. This analysis revealed that cells in both populations undergo oscillatory constrictions. Between constrictions, surface areas re-expand with gradually decreasing amplitude (Fig. 7a, e), which is most pronounced in MHP cells. The time of oscillation between two expansions revealed no significant difference between both groups (Fig. 7c, median oscillation time of 45 s). Likewise, there was no significant difference in the timing of individual constrictions. However, MHP cells spent less time expanding than MHP-adjacent cells (15 s median time for MHP vs 30 s for MHP-adjacent, Fig. 7d).

Together these data indicate that MHP cells and their neighbors undergo progressive narrowing of their apical pole via oscillatory constrictions and that these cellular dynamics proceed in a posterior-to-anterior direction.

**Neural fold fusion is initiated at closure points and mediated by protrusive activity.** In mice, neurulation proceeds unevenly along the anterior-posterior axis with multiple closure initiation sites[3]. To address whether NF fusion also occurs asynchronously in zebrafish, we performed time-lapse imaging of embryos mosaically expressing mGFP around the time when opposing NFs

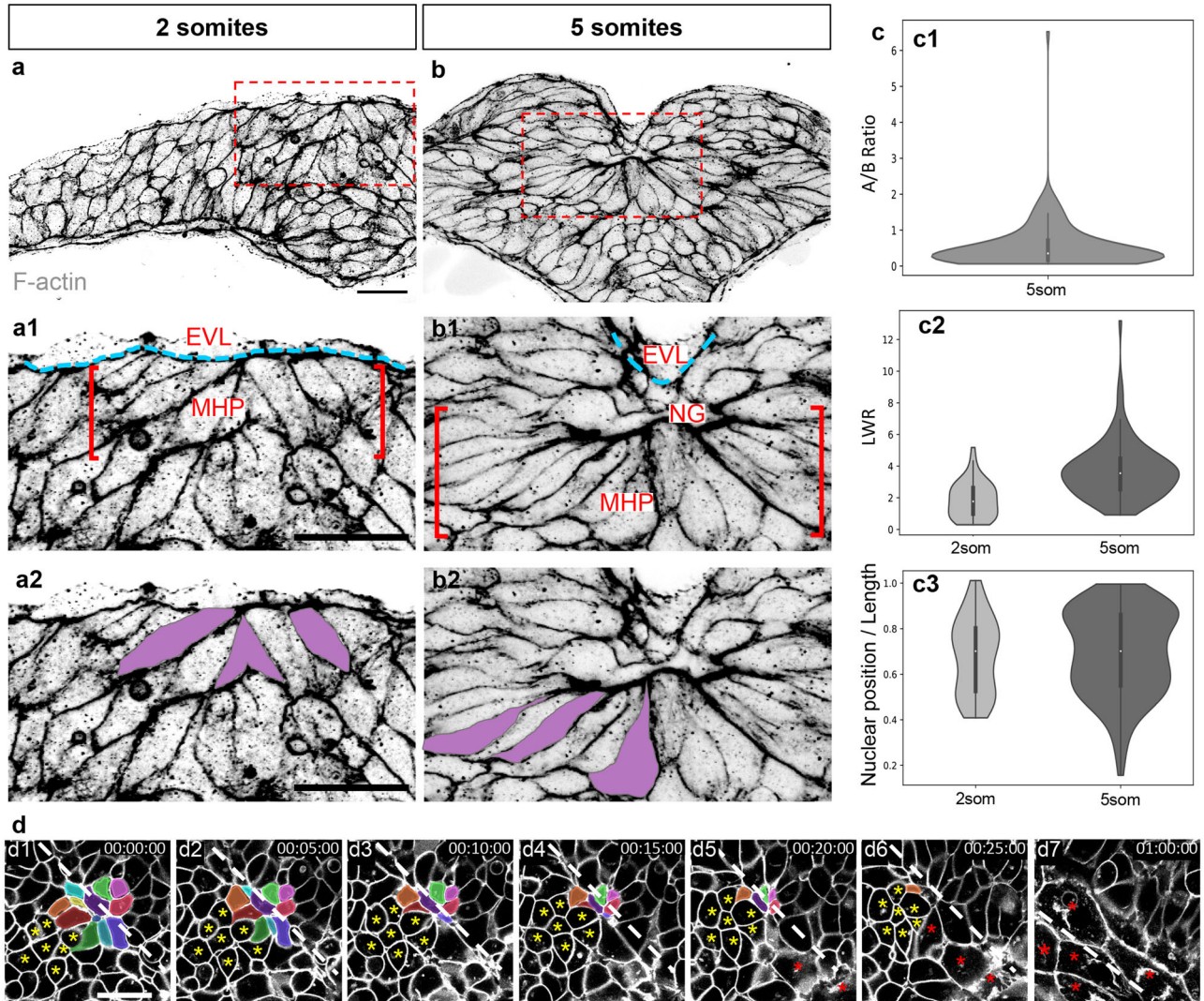

**Fig. 6 Apical constriction of MHP cells. a–b2** Transverse sections through the ANP at the 2 (**a**, **a1**, **a2**) and 5 (**b**, **b1**, **b2**) som stages labeled with phalloidin (shown in greyscale). **a1–b2** are higher magnifications of the boxed areas in **a** and **b**, revealing the organization of the medial ANP (**a1**, **b1**) and the shape of individual MHP cells pseudo-colored in purple (**a2**, **b2**). **c** Quantification of cell shape changes. Boxplot elements within the violin plots depict quartiles with the centerline depicting the median. **c1** Measurements of apical:basal surface ratio at 5 som ($n = 115$ cells from four embryos, mean = 0.548). **c2** Measurement of length-to-width (LWR) ratio at 2 som ($n = 47$ cells from 5 embryos, mean = 1.88) and 5 som (same cells as in **c1**, mean=3.70). A Mann–Whitney two-sided U Test revealed that the LWR increase between 2 som and 5 som is statistically significant ($P = 7.80e^{-11}$, AUC = 0.176). **c3** Relative position of nucleus at 2 and 5 som measured in the same cell populations (**c2**). Mean nuclear position/cell length (0.682 at 2 som vs 0.696 at 5 som) is not statistically significant using a Mann–Whitney U test ($P = 0.419$, AUC = 0.460). **d** Still frames of time-lapse movie of mGFP-labeled embryo imaged from a dorsal view. Individual MHP cells are pseudo-colored, a cluster of cells adjacent to the MHP is indicated with yellow asterisks and EVL cells are labeled with red asterisks. *EVL* enveloping layer, *MHP* medial hingepoint; *NG* neural groove. Annotations: white dashed line = midline, red brackets = MHP region, blue dashed lines = outlines *EVL*, yellow asterisks: cells adjacent to MHP, red asterisks: EVL cells. Scale bars: 25 µm in **a**, **a1**, and **a2**; 10 µm in **d1**.

approach the midline ($n = 2$ embryos, Fig. 8a and Supplemental Movie 6). Neuroectodermal cells of the NFs were identified in dorsal views based on their elongated shape and superficial location.

These movies revealed that the NFs have an arc shape with the apex positioned anteriorly (red asterisks in Fig. 8a), similar to the expression domain of the telencephalon marker *emx3* at the onset of NF convergence (inset in Fig. 8a). NF fusion is initiated near the apex of the arch and proceeds in an anterior-to-posterior direction for a distance of ~35 µm (red dotted line in Figure 8a3). At this level, defined as closure point one (C1 in Figure 8a3), NF fusion proceeds asynchronously as a second closure initiation site is formed more posteriorly (C2 in Figure 8a5). These closure

points define an eye-shaped opening (white dotted oval in Figure 8a4, a5) and a zippering process begins at the anterior and posterior corners of this structure, progressing from both ends toward the middle (Figure 8a6,a7; Supplemental Movie 6).

The events that complete NF fusion in amniotes involve the extension of dynamic cellular projections towards the midline[18–20]. Likewise, we observed that zebrafish neuroectodermal cells use filopodia to establish contact with cells from the contralateral side. Furthermore, the cell bodies appear to hyper-extend beyond the midline (Figure 8b3, b4 and Supplemental Movie 7), suggesting that they interdigitate between cells of the opposing NF. These cellular extensions are eventually retracted, as the midline becomes well defined after epithelialization (Fig. 2l).

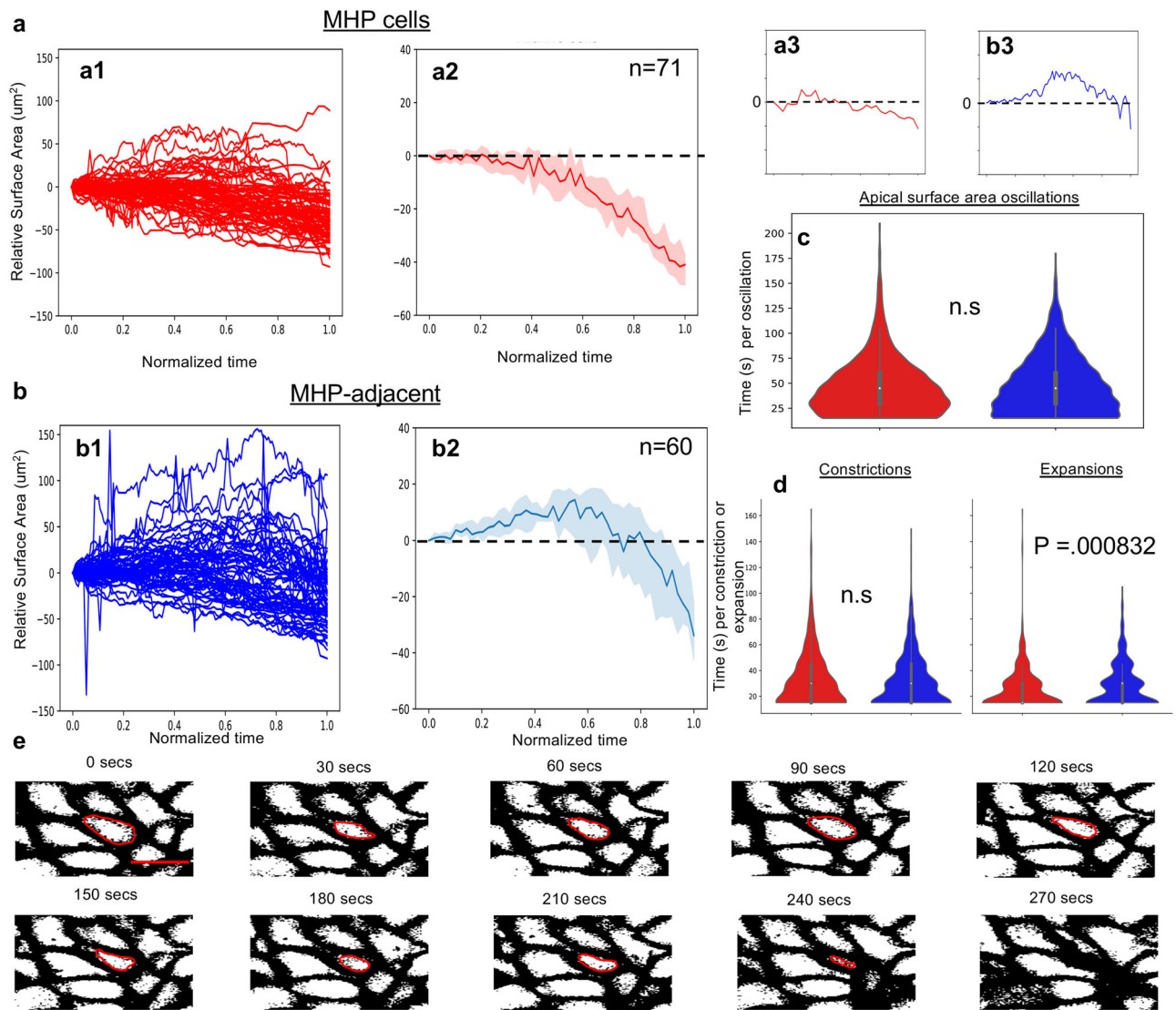

**Fig. 7 Oscillatory constriction with decreasing amplitude reduces the apical surface of MHP cells. a** Measurements of medial, MHP cells. **a1** Relative apical surface areas over time for individual MHP cells. **a2** Median values of MHP relative apical surface areas over time, 95% confidence interval, $n = 71$. **a3** Representative trace of relative apical surface area over time for an individual MHP cell. **b** Measurements of MHP-adjacent cells. **b1** Relative apical surface areas over time for individual MHP-adjacent cells. **b2** Median values of MHP-adjacent relative apical surface areas over time, 95% confidence interval. **b3** Representative trace of relative apical surface area over time for an individual MHP-adjacent cell. **c** Distributions of the duration of oscillation between two expanded states for MHP cells (red, median of 45 s per oscillation) and MHP-adjacent cells (blue, median of 45 s per oscillation). No significant difference (two-sided Mann–Whitney $U$ test, $P = 0.293$, AUC $= 0.489$, $n = 758$ MHP cell oscillations, $n = 1,113$ MHP-adjacent oscillations). **d** Distributions of the timing of apical constrictions or expansions for MHP cells (red) and MHP-adjacent cells (blue). There is no significant difference between the two groups for constriction time (two-sided Mann–Whitney $U$ test: constrictions, $P = 0.541$, AUC $= 0.510$, $n = 458$ MHP cell constrictions, $n = 640$ MHP-adjacent constrictions) but there is a statistically significant difference for expansion time ($P = 0.000832$, AUC $= 0.441$, $n = 367$ MHP cell expansions, $n = 596$ MHP-adjacent expansions). The median time for individual expansions of MHP and MHP-adjacent cells is 15 and 30 s, respectively. All boxplot elements depict quartiles with the centerline depicting the median. **e** Still frames of time-lapse movie of mGFP-labeled cells shown in greyscale. The oscillatory behavior of one cell, outlined in red, is shown over time. Scale bar: 10 μm in **e**.

The presence of closure points and the usage of filopodia to establish contact with NF cells across the midline reveal additional aspects of forebrain neurulation that are conserved in zebrafish.

**Temporal overlap in the timing of key cell shape changes in the ANP.** In amniotes, MHP formation precedes the shaping of paired DLHPs and initiates NF elevation. To reveal the relative timing of cell shape changes in zebrafish and their contributions to ANP morphogenesis, we plotted tissue dynamics using measurements

from transverse time-lapse movies ($n = 4$ embryos, data shown for one embryo). We observed that changes in the neural groove angle (indirect measurement of MHP apical constriction), the optic vesicle angle (which reflects DLHP cell elongation and apical constriction) and the basal NF angle (a proxy for epithelial kinking) occur concurrently and temporally correlate with NF elevation and convergence (Fig. 9). These analyses revealed that, in contrast to amniotes, key cell shape changes in the ANP do not occur in a distinct chronological manner.

We also measured the rate of these morphogenetic movements and observed that these values changed steadily over time ($n = 4$

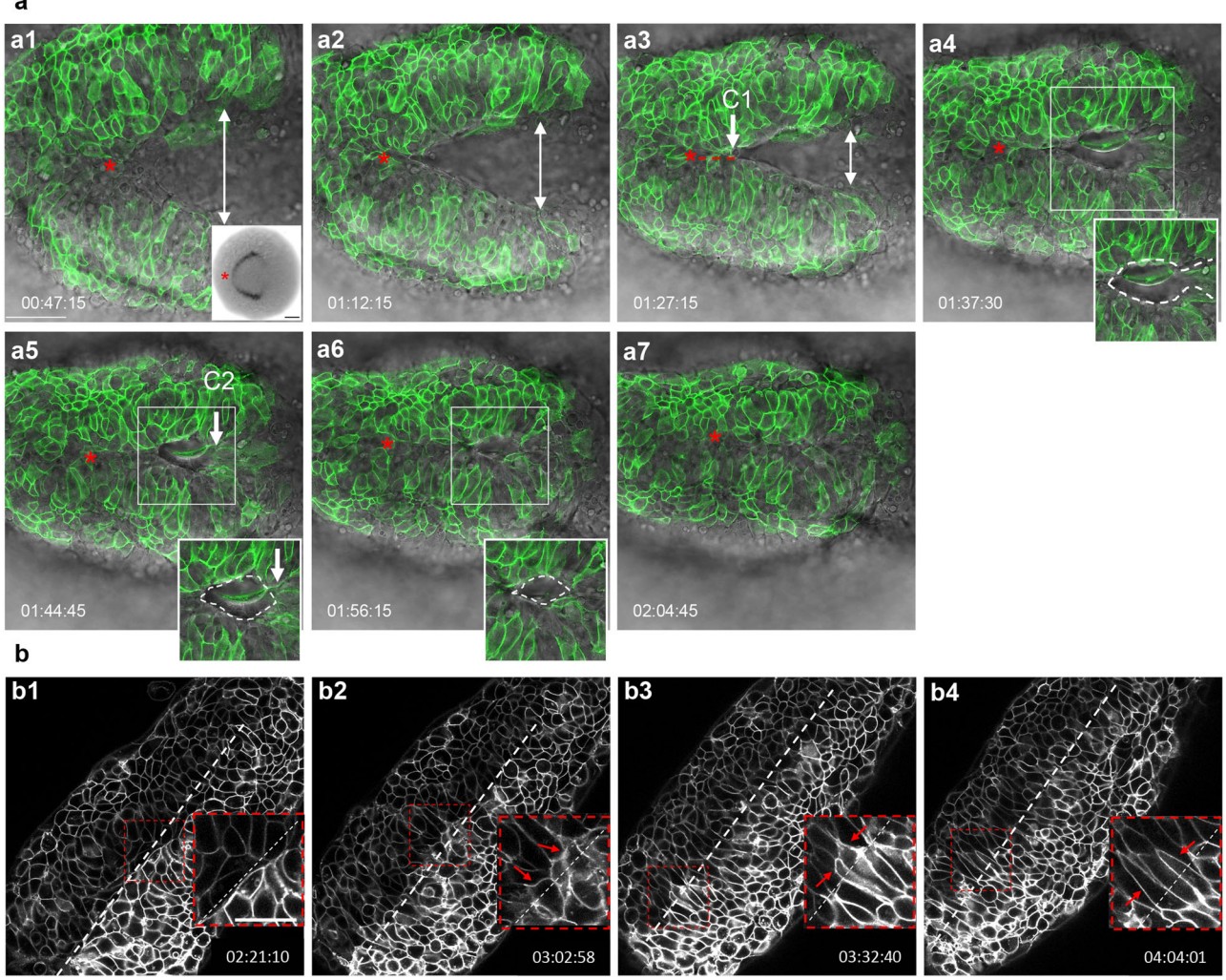

**Fig. 8 Dynamics of neural fold fusion. a**, **a1–a7** Time-lapse movie frames of an embryo expressing mosaic GFP, imaged from a dorsal view, showing the initiation of neural tube closure. Images are overlays of the green and brightfield channels. Inset in **a1** shows a dorsal view of an *emx3*-labeled embryo. Insets in **a4–a6** outline the eye-shaped opening that forms between closure sites one C1 and two (C2). **b** Greyscale time-lapse movie frames of an mGFP-labeled embryo imaged from a dorsal view, revealing the final stages of neural fold fusion. Insets in the lower right corner of **b1–b4** are higher magnification views of boxed areas. *C1, C2* closure sites one and two. Annotations: red asterisk: apex of the neural fold arc; red dotted line: synchronous and posteriorly-directed neural fold fusion anterior to closure point one; white double arrows: distance between the neural folds; white dotted oval: eye-shaped opening, the corners of which are defined by closure points one and two; white dotted line: embryonic midline; red arrows: filopodia extending across the midline; time-elapsed is shown at the bottom of each panel. Scale bars: 50 μm in **a1**, 100 μm in **a1** inset, and 25 μm in **b1**.

embryos, mean rates: NF elevation = 0.40 μm/min, NF convergence = −0.97 μm/min, neural groove angle = −2.193°/min, optic vesicle evagination = −1.73°/min and basal NF angle = −2.49°/ min).

**Molecular characteristics of medial and lateral hingepoints.** Key features of cells that form hingepoints include apico-basal polarization and accumulation of an apical contractile machinery composed of actin filaments (F-actin) and NMII[44]. To test whether the MHP and DLHPs in the zebrafish forebrain have some or all of these characteristics, the localization of apical markers Pard3-GFP (transiently expressed following mRNA injection), ZO-1 (anti-ZO-1) and N-cadherin (anti-N-cad) was examined along with F-actin (phalloidin) and phospho-Myosin Light Chain II (anti-P-MLC) in 5 som embryos (Fig. 10).

Apical co-localization of F-actin with Pard3-GFP (Figs. 10a–c2), ZO-1 (Figs. 10d–f2), and N-cad (Figs. 10g–i2) was confirmed in both the MHP and DLHPs at 5 som. Thus, although the

establishment of apico-basal polarity is generally delayed in the zebrafish neural plate relative to amniotes, the cell clusters that undergo apical constriction in the ANP epithelialize precociously. P-MLC accumulation at the apical pole is also apparent by the 5 som stage in MHP cells where it overlaps with F-actin (Figs. 10j-l2). In contrast, P-MLC apical enrichment is not observed in the DLHPs (arrowhead in Figs. 10k, l2). P-MLC does however accumulate in the cell cortex of neuroectodermal cells where it overlaps with F-actin, in addition to the basal surface of EVL cells (arrow in Figs. 10k, l1) and at the interface between the neuroectoderm and non-neural ectoderm (dotted line in Fig. 10k, l).

These findings confirm that both the MHP and DLHP undergo early epithelialization and accumulate apical markers. However, the MHP and DLHPs are molecularly distinct structures given that the latter is not enriched for P-MLC.

**Myosin contractility is required for MHP formation and neural fold convergence.** During neural tube closure, the

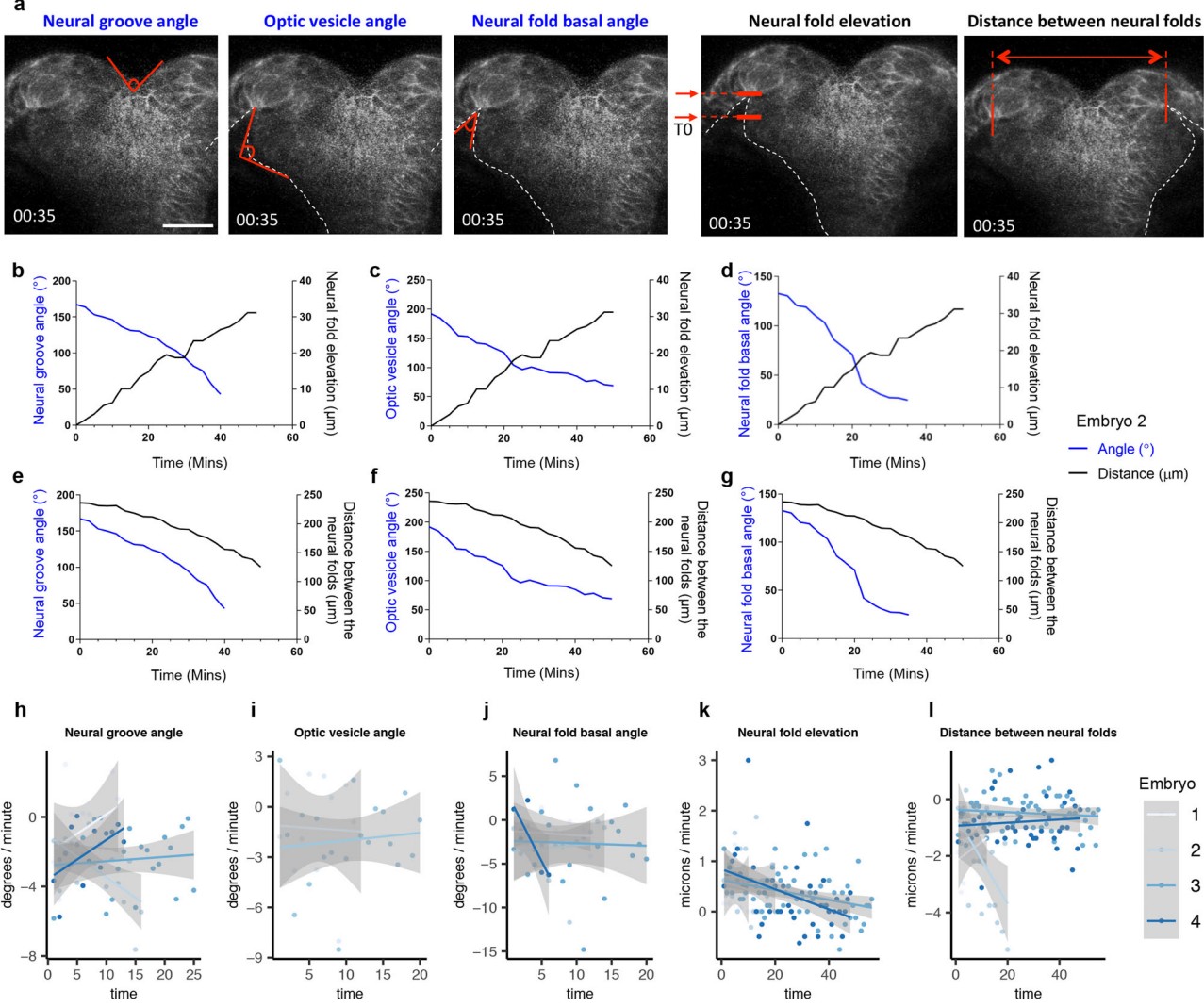

**Fig. 9 Dynamics of anterior neurulation. a** Still frames of an embryo expressing mKaede, showing how the measurements in graphs **b**–**l** were acquired, at a discrete time point. Neural groove was measured as the angle formed by the dorsal-most tissue; the optic vesicle angle was measured as the angle formed by the outline of the optic vesicle as it evaginates; the neural fold basal angle was measured as the angle formed by the basal surface of NF cells; the neural fold elevation was measured as the difference between the (elevated) position of the basal surface of NFs relative to its initial position at time zero (T0); the distance between the neural folds is the measure of the distance between the basal surface of NF cells. **b**–**g** Graphs illustrating the dynamics of neural groove formation (left Y axis, blue line in **b** and **e**), optic vesicle angle (left Y axis, blue line in **c** and **f**) and neural fold basal angle (left Y axis, blue line in **d** and **g**) as compared with neural fold elevation (right Y axis, black line in **b**–**d**) and distance between the neural folds (right Y axis, black line in **e**–**g**) over time (X axis). **h**–**l** Measurements indicated in **a** were converted into rates: (measurement frame 2−measurement frame 1)/time_step. Solid lines represent a fitted linear model for rate measurements of four embryos with the standard error as the shaded area. Scale bar in **a** = 50 μm.

actomyosin cytoskeleton is thought to be a driving force for apical constriction[41]. To address whether this molecular motor mediates apical constriction in zebrafish, NMII was blocked using blebbistatin and a translation-blocking morpholino (MO) targeting NMIIb[45] and the effect was analyzed in 5 som embryos that were labeled to reveal the localization of F-actin (phalloidin) and P-MLC (anti-P-MLC). Both treatments resulted in the absence of a clearly defined MHP (Figure 11a1, a1' vs 11b1, b1', c1 and c1'). LWR scores for these cells were close to 1 (Fig. 11d), indicative of cell rounding, which prevented measurement of their apical:basal surface ratio. In contrast, although the length of DLHPs in treated embryos was reduced (Fig. 11d), these cells retained their wedge shape (Figure 11a2, a2' vs b2, b2', c2, c2), with apical:basal surface ratios of <1 (Fig. 11e). These observations are consistent with the lack of apical P-MLC enrichment in DLHP cells and support a

conserved function for actomyosin in driving apical constriction of MHP cells.

In *Xenopus* embryos, disruption of actomyosin blocks apical constriction and convergent extension, causing severe neural tube defects[15]. To test whether myosin is similarly required for forebrain closure in zebrafish, embryos were treated with blebbistatin and labeled at 2, 5, and 7 som via in situ hybridization using the telencephalon marker *emx3*. Quantification of the width of the posterior-most *emx3* domain revealed that NF convergence was impaired in blebbistatin-treated embryos relative to controls (untreated and DMSO-treated) (Fig. 11f, g). Although we cannot rule out that earlier convergent extension defects in the ANP delay NF convergence, failure of MHP cells to undergo apical constriction is likely to contribute to this phenotype. Indeed, disruption of several other proteins implicated in apical constriction, including

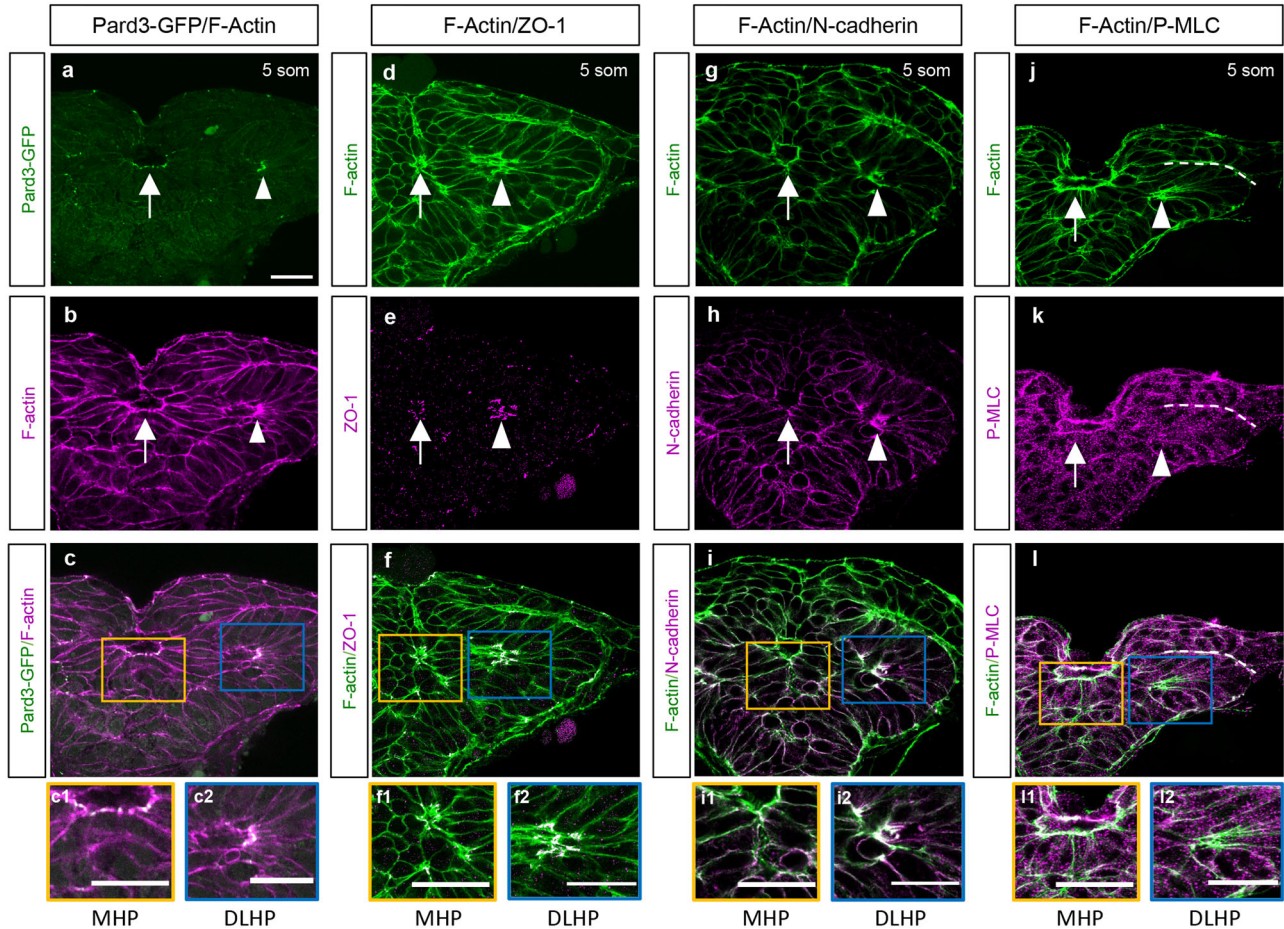

**Fig. 10 Molecular characterization of the MHP. a–l** Transverse sections through the ANP at the 5 som stage. Embryos double-labeled with Pard3-GFP (green) and phalloidin (F-actin, magenta) (**a–c2**); ZO-1 (magenta) and F-actin (green) (**d–f2**); N-cadherin (magenta) and F-actin (green) (**g–i2**); and with anti-P-MLC (magenta) and F-actin (green) (**j–l2**). **c, f, i, l** Magenta and green channel overlay. Insets show higher magnification images of the MHP (yellow boxes, **c1, f1, i1, l1**) and DLHP (blue box, **c2, f2, i2, l2**). Annotations: arrow = MHP; arrowhead = DLHP; dotted line in **j-l** = interface between the neuroectoderm and non-neural ectodermlayers of the neural folds; MHP medial hingepoint, DLHP dorsolateral hingepoint. Scale bars: 25 μm.

Shroom3[13] and GEF-H1, a RhoA-specific GEF[14] also causes neural tube closure defects.

## Discussion

We report here on mechanisms of forebrain morphogenesis in the zebrafish embryo and reveal that cell behaviors typically attributed to primary neurulation are observed in this brain region, namely, the use of hingepoints and NFs. These findings lend further credence to the view that primary neurulation is the ancestral condition[46].

The zebrafish MHP and paired DLHPs form in the superficial and deep layers of the eye field, respectively. The timing of MHP and DLHP formation overlap in zebrafish, in contrast to the biphasic nature of these events in mouse[3]. The zebrafish MHP is cup-shaped and more transient than its mammalian counterpart since these cells eventually intercalate between deep layer cells, contributing to the expansion of the eye vesicle[37]. Hingepoints are restricted to the ANP in zebrafish, however they are also present in more posterior regions in amniotes. Despite this difference, individual cells in the medial zone of the hindbrain neural plate were recently shown to internalize via a myosin-dependent mechanism[47]. Such variation from the organized cell clusters forming hingepoints in the zebrafish forebrain could be explained by precocious epithelialization of the ANP[37]. It thus appears that

there is a transition from clustered internalization in the forebrain to individual cell internalization in the hindbrain region.

A key feature of hingepoint cells is their reduced apical surface, which is in part owing to actomyosin contractility[13,41,48,49]. Apical constriction is thought to function as a purse string to generate the force required to bring the NFs together during cranial neurulation[50]. We provide evidence that the zebrafish MHP also utilizes an actomyosin-based contractile system. Assembly of this actomyosin network occurs via oscillatory constrictions with gradually decreasing amplitude, akin to the ratchet model initially proposed in invertebrates[51–53] and later reported during neural tube closure in *Xenopus*[54]. We further show that disruption of myosin impairs NF convergence, possibly by contributing a "pulling force" on the NFs or by clearing the dorsal midline of eye field cells. These observations suggest that the actomyosin machinery is used across vertebrates to drive cranial neural tube closure and it will be interesting in the future to test whether upstream regulators of apical constriction such as Shroom3[13] are also conserved.

In contrast to the MHP, the paired DLHPs do not require myosin to apically constrict, suggesting that DLHP formation is regulated by a distinct mechanism. Consistent with this observation, cell packing at a dorso-ventral boundary in the mouse neural tube causes buckling of the neuroectoderm at the DLHPs[55]. It is possible that such a mechanism also operates

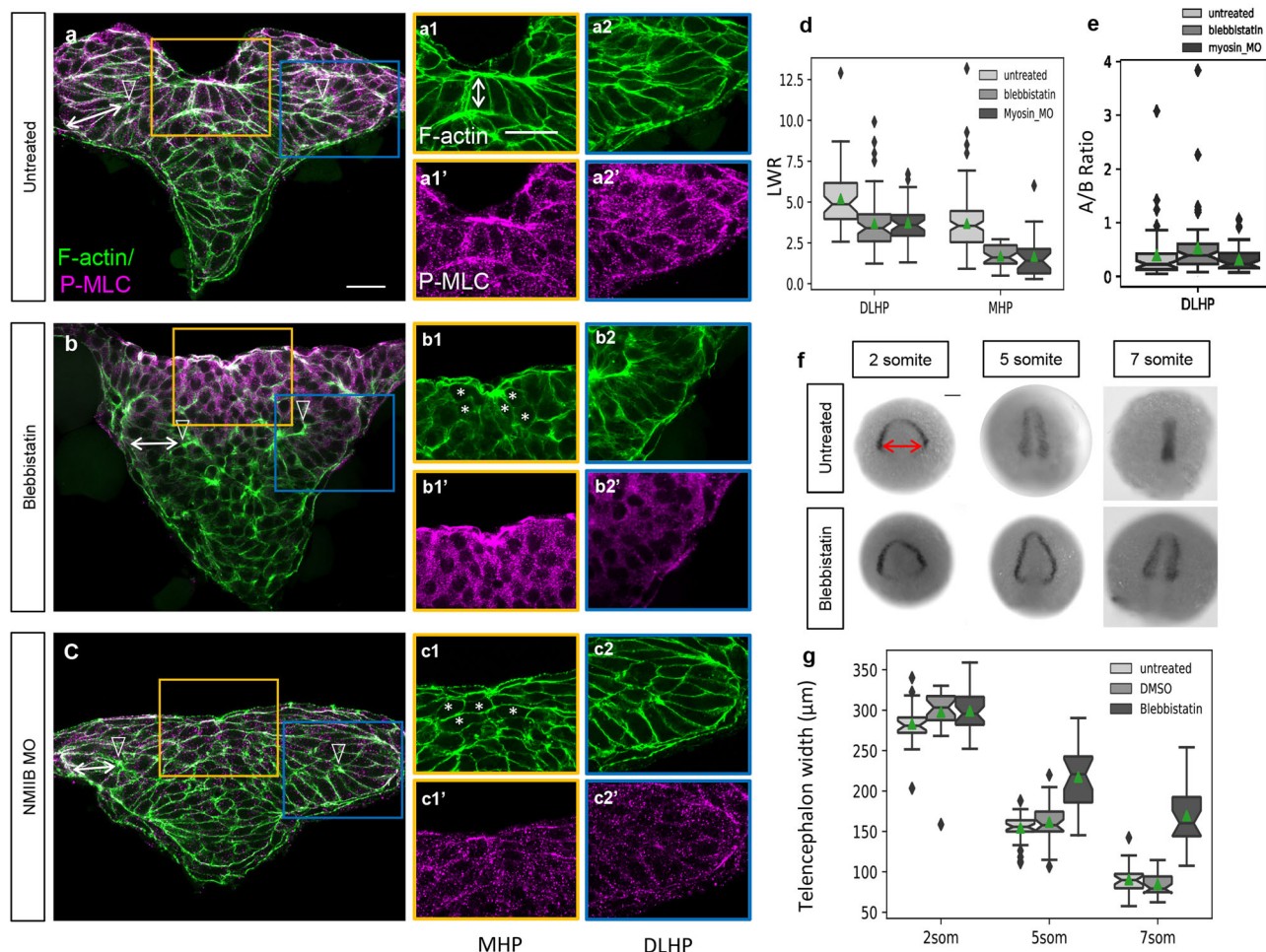

**Fig. 11 Role of non-muscle myosin II in apical constriction and neural fold convergence. a–c2′** Transverse sections through the ANP of 5 somuntreated (**a–a2′**), blebbistatin-treated (**b–b2′**) and NMIIB MO-injected embryos (**c–c2′**). Embryos were labeled with phalloidin (F-actin, green) and anti-P-MLC (magenta). **d** Length-width ratio distributions for DLHP and MHP cells from untreated, blebbistatin-treated, and NMIIB MO-injected embryos. DLHP measurements: untreated: $n = 50$ cells (five embryos), mean $= 5.22$; blebbistatin-treated: $n = 58$ cells (four embryos), mean $= 3.70$; NMIIB MO-injected: $n = 38$ cells (three embryos), mean $= 3.74$. MHP measurements: untreated: $n = 115$ cells (four embryos), mean $= 3.70$ (same data as in Fig. 6 c2); blebbistatin-treated: $n = 35$ cells (four embryos), mean $= 1.70$; NMIIB MO-injected: $n = 24$ cells (3 embryos), mean $= 1.68$. Two-sided Mann–Whitney $U$ test: DLHP: untreated vs blebbistatin: $P = 1.30e−6$, AUC $= 0.771$; untreated vs NMIIB MO-injected: $P = 2.14e−5$, AUC $= 0.766$; blebbistatin vs NMIIB MO-injected: $P = 0.325$, AUC $= 0.440$; MHP: untreated vs blebbistatin: $P = 3.55e−12$, AUC $= 0.889$; untreated vs NMIIB MO-injected: $P = 3.50e−8$, AUC $= 0.859$; blebbistatin vs NMIIB MO-injected: $P = 0.343$, AUC $= 0.574$. **e** Apical-to-basal length ratios of DLHP cells for untreated, blebbistatin-treated, and NMIIB MO-injected embryos, same cells quantified as in **d**. Two-sided Mann–Whitney $U$ test: untreated vs blebbistatin-treated $P = 0.00665$, AUC $= 0.348$; blebbistatin-treated vs NMIIB MO-injected: $P = 0.00827$, AUC $= 0.660$; untreated vs NMIIB MO-injected: $P = 0.930$, AUC $= 0.494$. **f** Dorsal views of 2, 5, and 7 som embryos untreated or blebbistatin-treated labeled via in situ hybridization using an *emx3* riboprobe. **g** Boxplots showing the distribution of telencephalon widths (double red arrow in **f**) according to treatment group. Notches depict the 95% confidence interval around the median and green triangles depict distribution means. 2 som: untreated: $n = 26$, mean $= 283.224$; DMSO-treated: $n = 26$, mean $= 297.727$; blebbistatin-treated: $n = 34$, mean $= 299.412$. 5 som: Untreated: $n = 33$, mean $= 154.688$; DMSO: $n = 30$, mean $= 161.747$; blebbistatin-treated: $n = 23$, mean $= 217.472$. Two-sided Mann–Whitney $U$ test: 5 somuntreated vs DMSO: $P = 0.332$, AUC $= 0.428$; 5 somuntreated vs blebbistatin-treated: $P = 1.30e−7$, AUC $= 0.0817$; 5 som DMSO-treated vs blebbistatin-treated: $P = 3.50e−6$, AUC $= 0.125$. 7 som: untreated: $n = 28$, mean $= 90.444$; DMSO-treated: $n = 27$, mean $84.855$; blebbistatin-treated: $n = 24$, mean $= 169.779$. Two-sided Mann–Whitney $U$ test: 7 somuntreated vs DMSO: $P = 0.170$, AUC $= 0.608$; 7 som untreated vs blebbistatin-treated: $P = 2.06e−9$, AUC $= 0.0140$; 7 som DMSO-treated vs blebbistatin-treated: $P = 1.46e−9$, AUC $= 0.00435$. Annotations: double white arrows $=$ cell length in deep layer; open arrowhead $=$ DLHP; asterisks $=$ rounded neuroectodermal cells; red double arrow $=$ posterior-most telencephalon width. Scale bars: 25 µm in **a** and **a1**; 100 µm in **f**.

during zebrafish neurulation. Actomyosin may however generate cortical tension[15], enabling both MHP and DLHP cells to maintain their elongated shape.

NFs in chick embryos form via epithelial ridging, kinking, delamination, and apposition[16], although the cellular basis of these morphogenetic events is not well understood. Elevation of the NFs in the mouse cranial neural plate is dependent on MHP constriction and expansion of the head mesenchyme[56]. NFs in the zebrafish are restricted to the ANP and head mesenchyme is unlikely to play a significant mechanical role in NF elevation, as the mesoderm layer underlying the ANP is thin. We instead identify narrowing of the basal surface of NF cells as a cell shape change that occurs concomitantly with elevation of the NFs. Based on scanning EM images of chick embryos, it appears that this "reverse hingepoint" structure

is also observed in amniotes and is linked to neural fold elevation or ridging/kinking[16,17]. It is tempting to speculate that narrowing of the basal surface of NF cells is driven by an active process such as basal constriction.

The final step of primary neurulation involves the convergence and fusion of the NFs at the dorsal midline. NF convergence in zebrafish is likely to be mediated by cell-intrinsic forces, including MHP and DLHP formation, although it is difficult to parse out their relative contribution, in addition to extrinsic forces derived from the non-neural ectoderm[17]. NF fusion is initiated at closure points in mammals and birds[57]. We observe two closure points in the zebrafish forebrain that form an eye-shaped opening that narrows from the corners in a bidirectional manner. Fusion of the NFs is mediated by the formation of cell protrusions that span the midline and establish the first points of contact with NF cells from the contralateral side. Akin to mice[21,22], the cells that initiate contact between apposing NFs in zebrafish derive from the neuroectodermal portion of the NFs. However, rather than establishing contact and adhering via their lateral surfaces, the protrusive ends of zebrafish neuroectoderm cells appear to transiently interdigitate between their contralateral counterparts, forming a rod-like structure, the precursor of the telencephalon. These hyperextensions are later retracted as the telencephalon eventually epithelializes, forming a clearly defined midline. Once the neuroectoderm cells have met and fused, non-neural ectoderm cells complete their migration and fuse at the dorsal midline, forming a continuous epidermal layer.

In addition to the epidermis, frog and fish embryos are surrounded by an outer protective epithelial monolayer, called EVL in zebrafish. Given that the EVL is indirect contact with the NFs, it is possible that this epithelial layer contributes to forebrain morphogenesis by providing a stable substrate for NF convergence. Conversely, the EVL is transiently tugged inward as the MHP cells undergo apical constriction. It is therefore likely that the interactions between the neural ectoderm and the EVL are reciprocal.

In summary, we reveal that zebrafish forebrain neurulation presents multiple similarities with primary neurulation but also some unique features. These findings contribute to our understanding of the evolution of neurulation and highlight the relevance of zebrafish to understand human neural tube development.

## Methods

**Zebrafish strains/husbandry**. Studies were performed using wildtype (AB) strains or Tg(emx3:YFP)[b1200] [39] embryos raised at 28.5ºC. Embryos from developmental stages 2–10 somites were used (~10–14 h post fertilization). The sex of the embryos used is unknown. All experiments were approved by the University of Maryland, Baltimore County's Institutional Animal Care and Use Committee (IACUC) and were performed according to national regulatory standards.

**Nucleic acid and MO injections**. Plasmids encoding membrane-targeted Green Fluorescent Protein (mGFP) (Richard Harland, University of California, Berkeley, CA, USA), membrane-targeted Red Fluorescent Protein (mRFP), membrane-targeted Kaede (mKaede) (a gift from Ajay Chtinis, National Institute of Health, Bethesda, MD) and pard3:egfp[58] were linearized with NotI and transcribed using the SP6 mMESSAGE mMACHINE kit (Ambion, AM1340). For ubiquitous expression of mGFP, mRFP, mKaede or Pard3:eGFP, 50 pg of RNA was injected into 1-cell stage embryos. For mosaic expression of mGFP, 50 pg of RNA was injected into 1 or 2 of the four central blastomeres at the 16-cell stage. These blastomeres have a high probability for neural fate[59] and are easy to identify for reproducible injections.

MOs were designed and synthesized by GeneTools (Philomath, Oregon, USA) and injected into one-cell stage embryos. mhy10 (NMIIB, EXON2-intron2) was delivered at 3 ng per injection.

mhy10: 5′-CTTCACAAATGTGGTCTTACCTTGA-3′[45]

Microinjections were performed using a PCI-100 microinjector (Harvard Apparatus, Holliston, MA, USA).

**Blebbistatin treatment**. 90% epiboly embryos were manually dechorionated in an agarose-covered petri dish with E3 medium. Once embryos reached the tailbud stage, they were placed into 50 μM blebbistatin (B0560 Sigma Aldrich) diluted with E3 and then incubated at 28.5 °C. Stock solution of blebbistatin was prepared with dimethyl sulfoxide (DMSO) as per the manufacturer's instructions. Accordingly, a control group of embryos were treated with 1% DMSO diluted with E3 alongside every blebbistatin trial. Once the desired developmental stage was reached (2-, 5-, or 7 som), embryos were immediately fixed with 4% paraformaldehyde (PFA). As blebbistatin is light-sensitive, embryos were kept in the dark as much as possible until fixation.

**Fixed tissue preparations and immunolabeling**. Embryos were fixed in 4% PFA for 16 h at 4 °C overnight. Immunolabeling was performed on whole-mount embryos, which were then sectioned with a Vibratome (Vibratome 1500 sectioning system), with the exception of anti-N-cadherin immunolabeling, which was done on sections. Primary antibody incubation was performed for 24–48 h at 4 °C and secondary antibody incubation for 2.5 h at room temperature.

Antibodies used: Rabbit anti-GFP at 1:1000 (Invitrogen, A11122), Rabbit anti-Sox3c at 1:2000 (Gift from Michael Klymkowsky), Rabbit anti-P-myosin light chain at 1:50 (Cell Signaling Technology, #3671 s), Mouse anti-p63 at 1:200 (Santa Cruz BioTechnology, SC-8431, no longer in production), Rabbit anti-N-cadherin at 1:50 (Abcam, ab211116) and Mouse anti-ZO-1 at 1:100 (Invitrogen, 33-9100). Alexa Fluorophore secondary antibodies were all used at a 1:1000 concentration: Goat anti-Rabbit −488, −568, −594 and Goat anti-Mouse −488, −594. Alexa Fluor 488-conjugated or 594-conjugated Phalloidin (Invitrogen, A12379 and A12381) at 1:250 and DAPI (Invitrogen, D1306) were used according to manufacturer's instructions. Sections were mounted on glass slides using ProLong Diamond Antifade Mountant (Invitrogen, P36961). Tg(emx3: YFP)[b1200] embryos were immunolabeled with anti-GFP to amplify the signal. mGFP and pard3:eGFP-injected embryos were not immunolabeled with anti-GFP.

**Whole-mount in situ hybridization and imaging**. In situ hybridization was performed as described[60]. Embryos were fixed in 4% PFA, dehydrated in methanol and rehydrated stepwise in methanol/PBS then 100% PBT (1× PBS 0.1% Tween 20). Embryos were incubated in hybridization buffer for 1 h followed by hybridization at 70 °C overnight. Following washes in hybridization buffer/SSC, embryos were incubated with preabsorbed alkaline-phosphatase coupled anti-digoxigenin antibody from Roche (Sigma aldrich, SKU 11093274910), at 1:5000 final dilution overnight at 4 °C. Detection was performed using an alkaline-phosphatase substrate solution (NBT, Sigma Aldrich SKU 11383213001; BCIP, Sigma Aldrich, SKU 11383221001). Reaction was stopped by washing in PBT.

emx3 riboprobe template was generated by PCR amplification using cDNA from 24hpf embryos.

T7 promoter: **TAATACGACTCACTATAGGG**

emx3 antisense:

FWD: TCCATCCATCCTTCCCCCTT

RVS: **TAATACGACTCACTATAGGG**GTGCTGACTGCCTTTCCTCT

DIG-labeled riboprobes were generated using 2ul of PCR template with the Roche DIG RNA Labeling Kit (T7) (Sigma aldrich, SKU 11277073910)

Whole-mount imaging was carried out using a Zeiss Axioscope2 microscope. Embryos were imaged in a 2.5% glycerol solution.

**Confocal microscopy**

*Dorsal view time-lapse microscopy*. Embryos were imaged using a Leica confocal microscope (Leica SP5 TCS 4D) at 15 s/frame capturing < .5μm of tissue. All fluorescently labeled sections were imaged using a Leica confocal microscope (Leica SP5 TCS 4D). Dechorionated live embryos were embedded in <1 mm holes bored in 1.2% low melting agarose in E3 medium, solidified on a glass-bottom dish with size 1.5 coverslips, as previously described[61]. Embryos were oriented such that the anterior side was pressed against the glass.

*Transverse view time-lapse microscopy*. Embryos were imaged using a Zeiss confocal microscope (Zeiss 900 LSM with Airyscan 2), at ×20. Dechorionated live embryos were embedded in 1× low melt agarose in E3 medium, in a glass-bottom dish with size 1.5 coverslips. Embryos were oriented such that the ventral side was pressed against the glass.

**Data quantification**

*Measurement of neural fold migration (NNE portion)*. Fig. 2n: For each half of a tissue section, the distance between the medial-most p63-positive nucleus and the midline was manually scored. Measurements were taken from tissue sections at different levels along the anterior-posterior axis of the forebrain.

*Deep layer cell morphology measurements*. Fig. 4: Cells were manually scored from non-projected z-stack images, where cellular outlines were visually determined from the mGFP signal. Cells were labeled as neuroectoderm NF cells based on the fan-like appearance of their basal projections in contact with the non-neural ectoderm. All other cells scored were within the morphologically distinct eye vesicle and were labeled as eye field cells. Neural ectoderm NF cells and eye field

cells are not morphologically distinguishable at the two somite stage and thus were not given cell type identities at that developmental stage.

*Medial hinge cell morphology measurements.* Fig. 6c: Cells were manually scored from non-projected z-stack images of the overlay between the labeled actin cytoskeleton (phalloidin) and nucleus (DAPI) channels. Cellular outlines were visually determined from the phalloidin signal. Figure 6c3: For the nuclear position measurement, the distance from the dorsal side of the nucleus to the basal membrane of the cell was scored and divided by the total cell length.

*Cell ratcheting measurements.* Fig. 7 a1,a2,b1,b2: Live movie z-stacks for the first 35 min (~2–4 somites, $n = 2$) of each movie were max projected and cropped so the midline of the tissue horizontally bisected the image frame. The mGFP signal was then inverted and thresholded to produce a binary image. Measurements for individual cell surface areas were captured using the magic wand tool in ImageJ for every frame (15 s) until the cell left the field of view. Plotted on the $x$ axis is the order of frame measurements (frame 0, frame 1, frame 2, etc.) divided by the total number of frames for which that cell was scored to standardize between 0 and 1. Plotted on the y axis, the initial value of each cell's surface area was subtracted from all measurements for that cell to initialize surface area's to zero. The data was binned (bins = 50) to obtain median measurements and confidence intervals. Cells that underwent mitosis anytime during the movie were excluded. Cells were labeled as MHP if their surface area centroid from the first frame ($t = 0$) was ±20 μm from the midline with MHP-adjacent cells being labeled as such if their centroid was >±0 μm from the midline.

*Measurement of oscillations using dorsal view movies of the apical surface.* An oscillation was defined as a cycle of constriction and then expansion or vice versa. For a given cell, constriction or expansion was determined by the difference between surface areas for two sequential time frames, where frame_1_surface_area - frame_2_surface_area >0 was scored as constricting and <0 was scored as expanding.

*Measurements of cell shape changes using transverse view movies.* Movies were manually scored at each time frame to measure the parameters illustrated in Fig. 9a ($n = 4$ embryos, data shown for one embryo only). Neural groove formation was calculated as the angle formed by the dorsal-most medial tissue; optic vesicle angle was measured as the angle formed by the outline of the optic vesicle as it forms; NF basal angle was calculated as the angle formed by the basal surface of NF cells; NF elevation was measured as the difference between the (elevated) position of the basal surface of the NFs compared to its initial position at time zero (T0); the distance between the NFs was measured as the distance between the basal side of NF cells, as they migrate towards the midline.

*Measurement of NF convergence.* Fig. 11c: the distance between the lateral edges of the posterior-most *emx3*-positive domain was manually scored for each embryo.

**Statistical analysis.** The Mann–Whitney $U$ test was used for all significance testing. The python function scipy.stats.mannwhitneyu(alternative = 'two-sided') was used to calculate the test statistic and $P$ value for each significance test. Graphs were generated using the python Seaborn package, with the exception of Fig. 9h–l, with the following functions: seaborn.boxplot(), seaborn.lmplot(), seaborn.violinplot(), seaborn.lineplot(). The panels in Fig. 9h-l were generated in R[62] (version 3.6.0) with ggplot2[63] (version 3.3.0) using the functions geom_point() and geom_smooth(method = 'lm').

**Statistics and reproducibility.** The non-parametric Mann–Whitney $U$ test was chosen for all significance testing to avoid making assumptions of normality for the underlying data. The effect size (AUC) for each reported $p$ value was calculated as $U_1/n_1*n_2$. Biological replicates in this study are defined as different embryos and all statistics (confidence intervals, $p$ values) were derived from measurements taken from at least three independent biological replicates (≥3 embryos), except for the analysis in Fig. 7, where the data analyzed are from independent live movies (two embryos).

**Reporting summary.** Further information on research design is available in the Nature Research Reporting Summary linked to this article.

## Data availability
Source data for the main figures can be found at Dryad[64] with the DOI 10.5061/dryad.ht76hrdd. The authors declare that any other data supporting the findings of this study are available within the article and its supplementary information files or from the corresponding author upon reasonable request.

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

## Acknowledgements

Funds from Howard Hughes Medical Institute through the UMBC Precollege and Undergraduate Science Education Program supported J. Werner and D. Brooks. Funds from NIH/NIGMS grants # T32-GM055036 and # R25-GM066706 and NSF LSAMP BD grant # 1500511 to UMBC supported M. Negesse. Funds from NSF LSAMP grant # 1619676 to UMBC supported J. Johnson. Funds from NIH/NIGMS MARCU*STAR T34 grant # HHS 00026 to UMBC supported D. Brooks and A. Caldwell. We thank the following people for their contributions: Tagide deCarvalho for her help with confocal imaging and image processing; Corinne Houart for the *Tg(emx3:YFP)*[b1200] transgenic line and her comments on the manuscript; Jennifer Gutzman for the gift of *myh10* MO and Mark Van Doren for his comments on the manuscript.

## Author contributions

J.W. and M.N. contributed equally to this study. J.W. pioneered this project, designed and performed experiments and carried out data analysis. M.N. carried out experiments and data analysis, annotated movie files and generated illustrations. D.B. and A.C. contributed to the experiments and analysis of cell polarity and myosin function. J.J. contributed to the analysis of myosin function. R.B. oversaw experimental design and analysis and wrote the manuscript.

## Competing interests

The authors declare no competing interests.
