## [Peer Review File · Communications Biology]

Reviewers' comments:

Reviewer #1 (Remarks to the Author):

The manuscript addresses the basic cellular rearrangements leading to neural keel/tube formation in zebrafish. The authors argue that certain processes, such as hinge point and neural fold formation, are evolutionary conserved between zebrafish and those observed in amphibians and amniotes, arguing that zebrafish (teleost) neurulation might be less unique than previously thought.

The data provided support their main claim of evolutionary conservation. My only and main concern is that the study as such is nearly entirely descriptive and that the descriptive tools do not go beyond previous attempts in the same direction. To argue for evolutionary conservation, not only morphogenetic aspects need to be analyzed but also mechanistic insight provided. There are a few functional experiments in the end of the study, but their validity - given the pleiotropic effects typically observed when using drugs - is questionable. My feeling is that this study as it stands now would be better suited for a more specialized journal, such as *Development*.

Reviewer #2 (Remarks to the Author):

Manuscript Number: COMMSBIO-19-1231-T

Hinge points and neural folds reveal conserved features of primary neurulation in the zebrafish forebrain

In this manuscript Werner et al., investigate the mechanism of zebrafish primary neurulation and show clearly that the zebrafish forebrain forms from the anterior neural plate based on the same cellular mechanisms that has previously been described in xenopus, chicken and mouse. This study further confirms previously developed hypotheses on conservation of neurulation features in vertebrates.

This is a very interesting and excellent piece of work. The manuscript is very well written, the data well investigated and presented. The authors use anatomical and molecular approaches to demonstrate the conservation of the mechanisms of primary neurulation between the zebrafish embryos, amniotes and amphibians.

My only critic concerns the figure legends in which the abbreviations are not always properly defined. For example, in figure 2 NF, M, NG and DL are not defined. MHP is defined although not mentioned on this figure. Please check all figure legends and correct them if necessary.

Reviewer #3 (Remarks to the Author):

The manuscript by Werner and colleagues investigates the cellular mechanisms underlying anterior neural tube formation in the zebrafish. By using cell labelling and confocal live imaging methods authors found that in the anterior portion of the neural tissue, neurulation occurs by the formation of two key morphological features of primary neurulation; namely a medial hinge point (MHP) and a pair

of dorso-lateral hinge points (DLHP). In addition, authors show evidence that myosin dependent apical constriction is required for MHP formation. While teleost neural tube formation has long been regarded to occur via cavitation rather than dorsal folding, the observations of Werner and colleagues suggest that in the anterior neural plate, zebrafish neurulation show reminiscent mechanisms found in primary neurulation (i.e. amniotes and amphibians).

The manuscript contains potentially interesting observations. However, due to lack of enough mechanistic insights in the present work there are several points of major criticism that need to be addressed before the manuscript is suitable for publication in *Communications Biology*.

1. The observation of MHP and DHLP in the anterior portion of zebrafish neural plate is interesting, but not sufficient to state that they are functionally equivalent to amniote primary neurulation. Authors need to show more convincingly live imaging data (in both transverse and anterior-posterior planes) regarding how cell and tissue rearrangements occur in the anterior portion of zebrafish neural anlage. Teleost forebrain development is a complex piece of morphogenesis and several authors have already shown that cell and tissue dynamics occur with a high degree of heterogeneity along both medial-lateral and anterior-posterior (Rembold et al., 2006; England et al., 2006; Ivanovitch et al., 2013). In this regard, authors should show evidence for the following points: 1) How does MHP contribute to DHLP formation, Can authors quantify neural groove formation and DHLP elevation?, To what extent these processes can be actually separated in a highly curved embryo? (having functional data from mutant conditions would be ideal), Why is MHP lost soon after tissue convergence and how does it contribute to the apical midline seam (lumen) development?; 2) How do lateral cells (prospective telencephalic progenitors) converge medially?, How much of this process is facilitated by eye field lateral expansion?; 3) How do cells fuse at dorsal side to allow closing?; 4) There is also very little description of how enveloping layer cells (EVL) dynamics might contribute to different steps of this process. Furthermore, while P63 marker has been used to label the basal layer of the developing epidermis, the superficial EVL is maintained as coherent tissue structure covering the early embryo during neurulation. So, how does EVL sheet dynamics relate to MHP/DHLP formation and dorsal closure?

2. While the observation that apical constriction is required for MHP formation is interesting, authors need to show more compelling evidence about apico-basal organization in the superficial neural plate layer. Most of the manuscript is written assuming a clear apico-basal organization of this tissue (i.e. p.6. "epithelial character"). While the early organization of the zebrafish neural plate is still a subject of debate, authors need to clarify with more strong evidence (i.e. apical markers of polarity such as Pard3, ZO-1 as well as other apical junctional components, alpha-cateinin and cadherins), to consolidate their major conclusions (especially at early time points of neurulation). As long as this analysis is not properly carried out I would suggest to authors to re-consider the term "epithelial" (widely used in the text) by using alternative definitions. Similar analysis should be performed for DHLPs.

3. The manuscript is fairly written in the light to propose anterior zebrafish neurulation as a case of primary neurulation. As long as authors can't convincingly demonstrate the above points I would suggest to authors to be more cautious about their conclusions as well as to keep the discussion more focused on the present mechanisms proposed for zebrafish neurulation, a developmental process still poorly explored.

Specific points

P6. Without having strong evidence for apico-basal organization of the plate I would suggest authors

to reconsider the title of this paragraph.

Figure 2. Can authors show if the morphological manifestation of MHP and DHLP can be completely ruled out from other regions within neural axis?, Can authors quantify DHLP?, Can authors introduce timing in their still images? (as well as in further figures).

P7. 140. Authors need to clarify whether they use "epithelial" versus "mesenchymal" to describe neural plate cells' character.

Figure 3. Additional live imaging is required to fully support these observations (see previous points). Why does MHP (neural groove) disappear at 7 somites?, and if so, does it occur?.

Figure 4. A. Authors should show cell organisation in rather lateral position (prospective DHLP). While authors mention the presence of cell protrusions during midline closing these features are not currently examined in detail.

Figure 6. The analysis of cell length change is not quite clear from the present images (a2 vs b2). In addition, I found the analysis of nuclear position slightly irrelevant as this data comes from still data (cross sections?).

Figure 7. Can the authors describe in the text the time of oscillations of MHP cells (i.e. pulses per minute?), How do authors choose the optical sections within the z-stack for this measurement? How many embryos/cells were analyzed? Does this oscillatory behaviour show any pattern? (anterior-posterior, medial-lateral)? Are these cell oscillations coordinated somehow across the tissue?

Figure 8. Authors should provide more convincing evidence of cellular rearrangements during midline sealing (i.e. using mosaic injection or transplanted cells), especially in the light to show the protrusive activity of neural cells. Furthermore, quantification is required. At present, this data is very hard to interpret.

Figure 9. Authors should characterize the molecular nature of the anterior neurlation in more depth (i.e. by using apico-basal markers as mentioned earlier). What does the arrow show in Fig 9A in relation with Pard3 expression?

Figure 10. What is the neural tube phenotype upon myosin disruption? While authors show an analysis of midline convergence (Fig. 10B) this does not necessarily reflect neural folds disruption. Authors need to show more clearly how MHP and DHLP are affected under these experimental conditions (quantification is highly desirable). Fig. 10a3, Control MO should be included.

Taken together, this is a potentially interesting manuscript, which need more decisive work before it is suitable for publication in Communications Biology.

RESPONSE TO REVIEWERS

We would like to thank the reviewers for their thoughtful and constructive comments on our manuscript entitled "Hingepoints and neural folds reveal conserved features of primary neurulation in the zebrafish forebrain". We feel that we have now addressed the main concerns, to the best of our ability, and believe that these changes significantly improve the quality of the manuscript. Please find below a point-by-point response to comments and concerns.

REVIEWER 1

The manuscript addresses the basic cellular rearrangements leading to neural keel/tube formation in zebrafish. The authors argue that certain processes, such as hingepoint and neural fold formation, are evolutionary conserved between zebrafish and those observed in amphibians and amniotes, arguing that zebrafish (teleost) neurulation might be less unique than previously thought.

Major Comments.

1. The data provided support their main claim of evolutionary conservation. My only and main concern is that the study as such is nearly entirely descriptive and that the descriptive tools do not go beyond previous attempts in the same direction.

Response. While we agree that these studies are mostly of a descriptive nature, we feel that this approach is warranted given that the end goal is to demonstrate conservation of cell behaviors whose molecular underpinnings are for the most part already understood. The novelty in the manuscript lies primarily in the correction of a long-held view that teleosts have evolved a unique mode of neurulation.

2. To argue for evolutionary conservation, not only morphogenetic aspects need to be analyzed but also mechanistic insight provided. There are a few functional experiments in the end of the study, but their validity - given the pleiotropic effects typically observed when using drugs - is questionable.

Response. The point regarding the pleiotropic effect of drugs is well taken. However, our rationale for using *Blebbistatin* is that it enabled us

Figure A. MHP formation is disrupted in *vangl2* mutants.

Transverse sections through the anterior neural plate of a 5 som wild type embryo (a1-a3) and *vangl2* mutant (a4-a6). Embryos are double-labeled with phalloidin (F-actin, green) and ZO-1 (magenta). a3, a6. Magenta and green channel overlay, with higher magnification of the MHP (yellow box) and DLHP (blue box) shown.

to control the timing of myosin inhibition. Temporal control is particularly important given that myosin is ubiquitous and required at multiple stages of development. We would further like to remind the reviewer that we also used morpholinos to disrupt non-muscle myosin IIb (NMIIB) and we have now included quantifications of cell shape changes (Figure 10).

We nevertheless attempted to address the reviewer's comment by examining the requirement of *vangl2*, a planar cell polarity (PCP) pathway component, for medial hinge point (MHP) formation. In *Xenopus* and mouse embryos this pathway has been implicated in multiple aspects of neural tube morphogenesis, namely convergent extension movements that narrow the neural plate and constriction of the MHP. We observed phenotypes in zebrafish *vangl2* mutants that are essentially consistent with these published observations (Figure A above), including impaired formation of the MHP (absence of F-actin and ZO-1 enrichment in medial superficial cells but not in the dorso-lateral domain where the DLHPs form). However, the morphology of the neural tube in these mutants is quite abnormal and is likely to contribute to the MHP defect we observe. In order to fully address the role of PCP signaling in MHP formation we would have to conditionally knock-out *vangl2*, which we feel extends beyond the already broad scope of this study. We have therefore chosen to not include the PCP data in our manuscript.

3. My feeling is that this study as it stands now would be better suited for a more specialized journal, such as *Development*.

Response. We hope that the insights and measurements obtained from transverse view time-lapse imaging and the extended analysis of cell polarity markers (requested by Reviewer 3) will mitigate this comment.

REVIEWER 2

In this manuscript Werner et al., investigate the mechanism of zebrafish primary neurulation and show clearly that the zebrafish forebrain forms from the anterior neural plate based on the same cellular mechanisms that has previously been described in *xenopus*, chicken and mouse. This study further confirms previously developed hypotheses on conservation of neurulation features in vertebrates.

This is a very interesting and excellent piece of work. The manuscript is very well written, the data well investigated and presented. The authors use anatomical and molecular approaches to demonstrate the conservation of the mechanisms of primary neurulation between the zebrafish embryos, amniotes and amphibians.

Minor Comment

1. My only critic concerns the figure legends in which the abbreviations are not always properly defined. For example, in figure 2 NF, M, NG and DL are not defined. MHP is defined although not mentioned on this figure. Please check all figure legends and correct them if necessary.

Response. We have now included the abbreviations in the figure legend for Figure 1 (previously Figure 2).

REVIEWER 3

The manuscript by Werner and colleagues investigates the cellular mechanisms underlying anterior neural tube formation in the zebrafish. By using cell labelling and confocal live imaging methods authors found that in the anterior portion of the neural tissue, neurulation occurs by the formation of

two key morphological features of primary neurulation; namely a medial hinge point (MHP) and a pair of dorso-lateral hinge points (DLHP). In addition, authors show evidence that myosin dependent apical constriction is required for MHP formation. While teleost neural tube formation has long been regarded to occur via cavitation rather than dorsal folding, the observations of Werner and colleagues suggest that in the anterior neural plate, zebrafish neurulation show reminiscent mechanisms found in primary neurulation (i.e. amniotes and amphibians).

The manuscript contains potentially interesting observations. However, due to lack of enough mechanistic insights in the present work there are several points of major criticism that need to be addressed before the manuscript is suitable for publication in Communications Biology.

Major Comments

1. The observation of MHP and DLHP in the anterior portion of zebrafish neural plate is interesting, but not sufficient to state that they are functionally equivalent to amniote primary neurulation. Authors need to show more convincingly live imaging data (in both transverse and anterior-posterior planes) regarding how cell and tissue rearrangements occur in the anterior portion of zebrafish neural anlage. Teleost forebrain development is a complex piece of morphogenesis and several authors have already shown that cell and tissue dynamics occur with a high degree of heterogeneity along both medial-lateral and anterior-posterior (Rembold et al., 2006; England et al., 2006; Ivanovitch et al., 2013).

Response. We now provide movies of time lapse transverse imaging to complement the anterior-posterior live imaging we performed previously (Supplemental Movies 1 and 4, Figure 4). These tissue dynamics are also shown and graphically represented in Figure 8.

We are familiar with the published work mentioned by this reviewer, which focuses on eye morphogenesis. Our cell-based analyses are in general agreement with the in toto imaging studies performed by Rembold et al. (2006) and England et al. (2006), which reveal patterns of cell movements. Ivanovitch et al. (2013) showed that the anterior neural plate undergoes early epithelialization and that superficial and medial cells of the eye field radially intercalate between cells in the deep layer. Overall, our findings build on these studies, by identifying hinge point structures and neural folds and examining the cellular dynamics that shape the forebrain and eye field as a whole. In addition, we provide evidence for a novel cell behavior, basal constriction of neural fold cells, that contributes to neural fold elevation. We have shown this in Figure 4 and updated the discussion on p19 I402.

2. In this regard, authors should show evidence for the following points:

2.1) 1) How does MHP contribute to DLHP formation, Can authors quantify neural groove formation and DLHP elevation? To what extent these processes can be actually separated in a highly curved embryo? (having functional data from mutant conditions would be ideal)

Response. We believe the reviewer meant to write “neural fold” elevation rather than “DLHP elevation”, as the latter is not thought to elevate. With respect to neural fold elevation, our analysis and quantification of transverse view time-lapse movies reveals that the timing of MHP formation (neural groove angle measurement) correlates with that of neural fold elevation (Figure 8) (p14 I301). That said, we cannot confirm causation. Thus, we tentatively conclude that MHP formation may contribute to neural fold elevation, possibly by clearing superficial eye field cells from the midline (p18 I386).

With respect to mutant analysis, we have examined $\text{vangl2}^{-/-}$ embryos in which planar cell polarity (PCP) signaling is disrupted (refer to response to Reviewer 1, comment # 2 and Figure A). PCP has been previously implicated in MHP formation and neural fold convergence in amniotes and we tentatively confirm this in zebrafish. However, the pleiotropic effect of vangl2 precludes drawing a strong conclusion in absence of additional experiments that extend beyond the scope of this paper. We have therefore chosen to not include the mutant analysis in this manuscript.

2.2) Why is MHP lost soon after tissue convergence and how does it contribute to the apical midline seam (lumen) development?

*Response. The zebrafish MHP is lost soon after tissue convergence as these cells undergo radial intercalation between deep cells of the eye field. These cellular dynamics were first reported by Ivanovitch et al. (2013), however this study did not identify the intercalating population as MHP cells. Using laser photoconversion of the fluorophore Kaede, we now confirm that the MHP cell population sinks inward, coinciding with the timing of neural fold convergence (Supplemental movie 1). This cell behavior is also observed in amphibians, albeit at a more posterior level, as superficial cells in the bilayered neural plate of *Xenopus* radially intercalate between underlying deep cells (Ossipova et al., 2015).*

2.3) How do lateral cells (prospective telencephalic progenitors) converge medially? How much of this process is facilitated by eye field lateral expansion?

Response. Based on our transverse view time-lapse movies (Supplemental Movies 1 and 4) it appears that there is a strong correlation between the timing of MHP (neural groove angle measurement) and DLHP (optic vesicle angle measurement) formation and neural fold convergence, but it has not been possible to parse out which of these processes is the main contributor, if any. Furthermore, it is likely that lateral forces from the non-neural ectoderm also play a role in neural fold convergence. We provide a general model for neural convergence in the Discussion section (p19 I408) that considers the contribution of both intrinsic and extrinsic forces.

2.4) How do cells fuse at dorsal side to allow closing?

Response. We previously addressed this question using dorsal view time-lapse imaging. This analysis revealed that neural fold fusion does not occur evenly along the anterior-posterior axis, but rather proceeds in a piecemeal manner initiated at closure sites (Figure 7A), as has also been reported in mouse and chick embryos. Following this initial phase, neural fold cells extend medially directed protrusions and elongate, intercalating between neural fold (prospective telencephalon) cells on the contralateral side, a process that has previously been coined interdigitation (Figure 7B). These hyperextensions are eventually retracted since we observe a clear midline after epithelialization has occurred (Figure 1L). The late aspect of neural fold fusion involving interdigitation is likely to be unique to teleosts as mammalian neural fold cells, although protrusive, fuse via contact between their lateral surfaces. We have now included a more extensive discussion of neural fold fusion in the Discussion section (p19 I416).

2.5) There is also very little description of how enveloping layer cells (EVL) dynamics might contribute to different steps of this process. Furthermore, while P63 marker has been used to label the basal layer of the developing epidermis, the superficial EVL is maintained as coherent tissue structure covering the early embryo during neurulation. So, how does EVL sheet dynamics relate to MHP/DHLP formation and dorsal closure?

Response. The reviewer raises an excellent point, unfortunately it has not been possible to directly assess the role of the EVL in morphogenesis, as our attempts to chemically remove it using Trypsin invariably resulted in embryonic lethality. We nevertheless acknowledge in the Discussion section (p20 I423) that the EVL may contribute to forebrain morphogenesis in teleosts by providing a stable substrate for prospective telencephalon cells to migrate on. Conversely, the EVL is transiently tugged inward as the MHP cells apically constrict (Figure 5B, b1). It is therefore likely that the interactions between the neural ectoderm and the EVL are reciprocal.

3. While the observation that apical constriction is required for MHP formation is interesting, authors need to show more compelling evidence about apico-basal organization in the superficial neural plate layer. Most of the manuscript is written assuming a clear apico-basal organization of this tissue (i.e. p.6. “epithelial character”). While the early organization of the zebrafish neural plate is still a subject of debate, authors need to clarify with more strong evidence (i.e. apical markers of polarity such as Pard3, ZO-1 as well as other apical junctional components, alpha-cateinin and cadherins), to consolidate their major conclusions (especially at early time points of neurulation). As long as this analysis is not properly carried out I would suggest to authors to re-consider the term “epithelial” (widely used in the text) by using alternative definitions. Similar analysis should be performed for DHLPs.

Response. We have now extended the initial analysis of F-actin and pard3-GFP to include Z01 (Figure 9 D-F) and N-cadherin (Figure 9 G-I). Importantly, all of these markers are enriched on the apical pole of MHP and DLHP cells (p15 /321). These findings corroborate with those of Ivanovitch et al. (2013) who were the first to describe early epithelialization of the anterior neural plate, even though they did not report on hinge point structures.

4. The manuscript is written in the light to propose anterior zebrafish neurulation as a case of primary neurulation. As long as authors can't convincingly demonstrate the above points I would suggest to authors to be more cautious about their conclusions as well as to keep the discussion more focused on the present mechanisms proposed for zebrafish neurulation, a developmental process still poorly explored.

Response. Overall, we feel that our data support conservation of key cellular mechanisms of neurulation in teleosts, however we acknowledge that there are also some interesting differences (potential role of the EVL, multilayered organization of the neural plate, mode of neural fold fusion), which we have included in several sections of the Discussion (p17/367, p19/416, p20/423, p20/429).

Minor Comments

1. P6. Without having strong evidence for apico-basal organization of the plate I would suggest authors to reconsider the title of this paragraph. [first paragraph of the results section].

Response. This paragraph directly builds on a study published by Ivanovitch et al (2013) who demonstrated that the anterior neural plate undergoes precocious epithelialization and should therefore not be misleading. Nevertheless, we have changed the title to more closely reflect the state of knowledge in the field at the time we began our study: “Precocious epithelialization of the anterior neural plate is associated with bending” (p6 I98). Towards the end of the Results section (p15 I314) we revisit the epithelial character of ANP cells and go on to demonstrate, using a battery of markers, that the eye field cells organize in clusters that function as hinge points (Figures 9 and 10).

2. Figure 2. Can authors show if the morphological manifestation of MHP and DHLP can be completely ruled out from other regions within the neural axis?

Response. We have examined cross sections at the level of the midbrain, hindbrain and spinal cord and never observed the morphological manifestations of a MHP or DLHPs. That said, Araya et al. (2019) did report myosin-dependent internalization of individual neuroepithelial cells in more posterior regions of the zebrafish embryo, indicating that different mechanisms are used to shape the neural plate along the anterior-posterior axis.

3. Can authors quantify DHLPs? Can authors introduce timing in their still images? (as well as in further figures).

Response. We previously quantified the length and apico-basal surface ratio of DLHPs in Figure 3 (lateral eye field cells). In this edited version we have also performed optic vesicle angle measurements (an indirect measure of apical constriction of the DLHPs) in WT embryos (Figure 8B, E, H). We further measure the impact of myosin disruption on DLHP formation in Figure 10.

All panels that show still frames of movies now have the time indicated

4. P7. 140. Authors need to clarify whether they use “epithelial” versus “mesenchymal” to describe neural plate cells’ character.

Response. To remove any confusion, we refer to the precocious epithelialization first reported by Ivanovitch et al. (2013) (p6 I98; p7 I134) and then go on to discuss how the distribution of F-actin corroborates with this finding.

5. Figure 3. Additional live imaging is required to fully support these observations (see previous points). Why does MHP (neural groove) disappear at 7 somites?, and if so, how does it occur?.

Response. We have now performed transverse view time-lapse imaging of embryos in which MHP cells were labeled via photo-activation of Kaede in the superficial/medial region of the eye field (Supplemental Movie 1). We observed that MHP cells sink inward around the time that the neural folds converge medially. These observations are consistent with the Ivanovitch et al. (2013) study that showed that superficial and medial eye field cells radially intercalate between cells in the deep layer. In this regard, the MHPs of zebrafish and Xenopus are different than their mammalian counterpart, which forms ventrally and remains in place. This difference reflects the multi-layered nature of the neural plate in teleosts and amphibians (which resolves overtime into a single-cell layered neural tube) relative to the pseudostratified neural plate of amniotes. Please refer to the second paragraph of the Discussion section (p17 I367).

6. Figure 4. A. Authors should show cell organization in rather lateral position (prospective DHLP). While authors mention the presence of cell protrusions during midline closing these features are not currently examined in detail.

Response. Cell organization in the lateral region of the neural plate is already shown at high resolution in Figure 2 (DLHP and neural folds) and Supplemental Figures 2 and 3. Furthermore, cell protrusions were capture in time-lapse movies (Supplemental Movie 7) and still frames (insets in Figure 7b1-4).

7. Figure 6. The analysis of cell length change is not quite clear from the present images (a2 vs b2). In addition, I found the analysis of nuclear position slightly irrelevant as this data comes from still data (cross sections?).

Response. The confusion most likely stems from the use of different scale bars for the images in panels a2 and b2. We now provide images shown at the same magnification (Figure 5).

Unlike in amniotes, the nucleus of zebrafish neuroectodermal cells does not change position in a cell cycle-dependent manner. We therefore think it is appropriate to score its relative position at different stages of development. For the data we compiled, there is no significant difference in nuclear position at 2 and 5 somites. This indicates that, in contrast to amniotes, nuclear repositioning is unlikely to be a mechanism for facilitating apical constriction and elongation of MHP cells, which we believe is an interesting species-specific difference worth mentioning (p12 l239).

8. Figure 7. Can the authors describe in the text the time of oscillations of MHP cells (i.e. pulses per minute ?), How do authors choose the optical sections within the z-stack for this measurement? How many embryos/cells were analyzed? Does this oscillatory behaviour show any pattern? (anterior-posterior, medial-lateral)? Are these cell oscillations coordinated somehow across the tissue?

Response. The focal plane was set immediately below the enveloping layer, at the level of the MHP. For the two movies used in this analysis (Supplemental Movie 5 and an additional movie not included in the supplemental material), the entire z-stack was max-projected prior to quantification. Z-depth of Supp. Movie 5: .3953 microns and z-depth of the additional movie: .1000 microns. Two embryos were analyzed, each comprising a separate movie. A combined total across both embryos of 71 medial cells and 61 marginal cells were quantified. The average length of each oscillation is ~ 45 seconds for both the medial and medio-lateral cells. The only pattern of oscillation apparent to us, as described in the text, is that medial cells spent less time expanding compared to medio-lateral cells. These details are now featured in the Figure legend of Figure 6 and in the results section (p13 l257).

9. Figure 8. Authors should provide more convincing evidence of cellular rearrangements during midline sealing (i.e. using mosaic injection or transplanted cells), especially to show the protrusive activity of neural cells. Furthermore, quantification is required. At present, this data is very hard to interpret.

Response. We agree with the reviewer that the analysis of protrusive activity and cellular rearrangements in the neural folds does not allow for strong conclusions about the dynamics of these events, however the focus of this paper was on the documentation of internal tissue structures (hinge points and neural folds). The analyses required to fully document protrusion dynamics and the steps leading to establishment of a defined apical midline in the neural tube would be extensive, as exemplified by an entire publication focused on a similar topic in the posterior neural plate of the zebrafish embryo (Buckley et al., 2012). We have nevertheless modified the Results section, to provide more conservative data interpretation (p14, l293-294).

10. Figure 9. Authors should characterize the molecular nature of the anterior neurulation in more depth (i.e. by using apico-basal markers as mentioned earlier). What does the arrow show in Fig 9A in relation with Pard3 expression?

Response. Our extended analysis of apico-basal markers should address this concern. The arrow in Figure 9A points to the apical pole of MHP cells, where Pard3-GFP is enriched. The arrowhead similarly reveals apical concentration of Pard3-GFP in one of the DLHPs.

Figure 10. What is the neural tube phenotype upon myosin disruption? While authors show an analysis of midline convergence (Fig. 10B) this does not necessarily reflect neural folds disruption. Authors need to show more clearly how MHP and DHLP are affected under these experimental conditions (quantification is highly desirable). Figure 10a3, Control MO should be included.

Response. We now provide higher magnification images of MHP and DLHP cells in Figure 10A-c2', along with measurements of length-to-width ratios in Figure 10D (previously figure 10B) and apical:basal surface ratios in Figure 10E. We further updated the results section to reflect these quantifications (p16 l342) as well as the discussion section (p18 l391). These data reveal that while MHP cells round up following blebbistatin treatment or NMM II morpholino injection, DLHP cells retain their wedge shape (consistent with the lack of apical enrichment of P-MLC (phospho-myosin light chain). While we did not include a morpholino control in this figure, the NMM II MO was validated by others (Gutzman et al. 2015) and we have controlled for the impact of the injection itself with mGFP mosaic expression, which does not cause cell rounding.

UPDATED FIGURES

Figure 1. Hingepoints and neural folds contribute to forebrain morphogenesis. (A-D) Optical sections at the level of the forebrain of WT embryos at the 2-3 som (A), 5 som (B), 7 som (C) and 10 som (D) stages. (E-L) Transverse sections through the ANP of 2-3 som (E,I), 5 som (F,J), 7 som (G, K) and 10 som (H, L) embryos. (E-H) Tg[emx3:YFP] embryos labeled with anti-GFP (green), phalloidin (F-actin, magenta) and DAPI (nuclei, blue). (I-L) WT embryos labeled with phalloidin (F-actin, green), anti-Sox3 (magenta) and anti-p63 (nuclear label, green). (M) Higher magnification image of panel J, grey scaled to reveal F-actin and p63 and pseudo-colored - color code: light blue: superficial eye field cells, dark blue: deep layer cells that apically constrict to form the optic vesicles, green: neural component of the neural fold, orange: olfactory placode (Sox3/p63-negative cells), yellow: non-neural component of the neural fold. (N) Measurements of neural fold convergence, scored as the distance between the medial-most p63-positive cells on either side of the midline at different developmental stages. Notches depict the 95% confidence interval around the median and the green triangle depicts the distribution mean. 2 som: 48 measurements from 14 embryos, mean= 136; 5 som: 144 measurements from 16 embryos, mean=73.0; 7 som: 87 measurements from 10 embryos, mean=36.0; 10 som: p63 domain is fused, no measurements. Statistical analysis: Mann-Whitney U tests, two-sided; 2 som vs 5 som : $P = 1.18e^{-21}$; 2 som vs 7 som : $P = 8.35e^{-22}$; 5 som vs 7 som : $P = 2.14e^{-31}$. (O) Side view of a 4 som

Tg[*emx3:YFP*] embryo. Abbreviations: s = superficial layer; d = deep layer; NF = neural fold; NG = neural groove; hyp = hypothalamus; tel = telencephalon; ov = optic vesicle; DL = Dorso-lateral hingepoints; M = Medial hingepoint; OP = olfactory placode; NE = neural ectoderm; NNE = non-neural ectoderm. Annotations: black arrowhead = median groove, white open arrowhead = elevated neural fold-like structure, dashed line = separation of the deep and superficial layers; brackets = hingepoints; dotted line = A-P range of the neural folds; white arrowhead = medial-most epidermis; red asterisk = neuropore. Scale bars: A and O= 100 μ m, E= 25 μ m.

Figure 4. Dynamics of neural fold formation. A-F. Time lapse movie frames of the ANP, from a transverse view. A-C. Still frames of an embryo expressing membrane Kaede (mKaede). D-F. Still frames of a Tg[*emx3:YFP*] embryo expressing membrane RFP (mRFP, pseudo labeled magenta) and YFP (green). Yellow boxes in A, C, D and F identify magnified areas in a1, c1, d1 and f1. Annotations: blue dotted line: outlines the basal side of the neural folds; red and white lines identify individual neural fold cells; arrows: indicate narrowing surface in neural folds cells. Scale bars: 50 μ m in A and D; 25 μ m in a1 and d1.

Figure 5. Apical constriction of MHP cells. (A-b2) Transverse sections through the ANP at the 2 (A, a1, a2) and 5 (B, b1, b2) som stages labeled with phalloidin (shown in greyscale). (a1-b2) are higher magnifications of the boxed areas in A and B, revealing the organization of the medial ANP (a1, b1) and the shape of individual MHP cells pseudo-colored in blue (a2, b2). (C) Quantitation of cell shape changes. Boxplot elements depict quartiles with the centerline depicting the median. (c1) Measurements of apical:basal surface ratio at 5 som (n= 115 cells from 4 embryos, mean=0.548). (c2) Measurement of length-to-width (LWR) ratio at 2 som (n= 47 cells from 5 embryos, mean=1.88) and 5 som (same cells as in c1, mean=3.70). A Mann-Whitney two-sided U Test revealed that the LWR increase between 2 som and 5 som is statistically significant ($P = 7.80e^{-11}$). (c3) Relative position of nucleus at 2 and 5 som measured in the same cell populations (c2). Mean nuclear position/cell length (0.682 at 2 som vs 0.696 at 5 som) is not statistically significant using a Mann Whitney U test ($P=0.419$). (D) Still frames of time lapse movie of m-GFP labeled embryo imaged from a dorsal view. Individual MHP cells are pseudo-colored, A cluster of cells adjacent to the MHP is indicated with yellow asterisks and EVL cells are labeled with red asterisks. Abbreviations: EVL = enveloping layer; MHP = medial hinge point; NG = neural groove. Annotations: white dashed line= midline, red brackets = MHP region, blue dashed lines = outlines EVL, yellow asterisks: cells adjacent to MHP, red asterisks: EVL cells. Scale bars: 25 μm in A, a1 and a2; 10 μm in d1.

Figure 6. Oscillatory constriction with decreasing amplitude reduces the apical surface of MHP cells. (A) Measurements of medial, MHP cells. (a1) Relative apical surface areas over time for individual MHP cells. (a2) Median values of MHP relative apical surface areas over time, 95% confidence interval, $n=71$. (a3) Representative trace of relative apical surface area over time for an individual MHP cell. (B) Measurements of MHP-adjacent cells. (b1) Relative apical surface areas over time for individual MHP-adjacent cells. (b2) Median values of MHP-adjacent relative apical surface areas over time, 95% confidence interval. (b3) Representative trace of relative apical surface area over time for an individual MHP-adjacent cell. (C) Distributions of the duration of oscillation between two expanded states for MHP cells (red, median of 45 seconds per oscillation) and MHP-adjacent cells (blue, median of 45 seconds per oscillation). No significant difference (two-sided Mann Whitney U test, $P=0.293$, $n=758$ MHP cell oscillations, $n=1,113$ MHP-adjacent oscillations). (D) Distributions of the timing of apical constrictions or expansions for MHP cells (red) and MHP-adjacent cells (blue). There is no significant difference between the two groups for constriction time (two-sided Mann Whitney U test: constrictions, $P=0.541$, $n=458$ MHP cell constrictions, $n=640$ MHP-adjacent constrictions) but there is a statistically significant difference for expansion time ($P = 0.000832$, $n=367$ MHP-cell expansions, $n=596$ MHP-adjacent expansions). The median time for individual expansions of MHP and MHP-adjacent cells is 15 and 30 seconds respectively. All boxplot elements depict quartiles with the centerline depicting the median. (E) Still frames of time-lapse movie of m-GFP labeled cells shown in grey-scale. The oscillatory behavior of one cell, outlined in red, is shown over time. Scale bar: 10 μm in c1.

Figure 8. Dynamics of anterior neurulation. A-F. Graphs illustrating the dynamics of neural groove formation (left Y axis, blue line in A and D), optic vesicle angle (left Y axis, blue line in B and E) and neural fold basal angle (left Y axis, blue line in C and F) as compared to neural fold elevation (right Y axis, black line in A-C) and distance between the neural folds (right Y axis, black line in D-F) over time (X axis). G-K. Still frames of an embryo expressing mKaede (green), in which the MHP cells were photoconverted (magenta), showing how the measurements in graphs A-F were acquired, at two discrete time points. G. The neural groove was measured as the angle formed by the dorsal most tissue. H. The optic vesicle angle was measured as the angle formed by the outline of the optic vesicle as it forms. I. The neural fold basal angle was calculated as the angle formed by the basal side of neural fold cells. J. Neural fold elevation was measured as the distance between the basal side of neural fold cells and the apical side of MHP cells. As the neural folds elevate, that distance decreases and eventually becomes negative as the neural folds elevate above the MHP. K. Distance between the neural folds was measured as the distance between the basal side of neural fold cells. Scale bar: 50 μm.

Figure 9. Molecular characterization of the MHP. (A-L). Transverse sections through the ANP at the 5som stage. Embryos double-labeled with Pard3-GFP (green) and phalloidin (F-actin, magenta) (A-c2); ZO-1 (magenta) and F-actin (green) (D-f2); N-cadherin (magenta) and F-actin (green) (G-i2); and with anti-P-MLC (magenta) and F-actin (green) (J-l2). (C, F, I, L) Magenta and green channel overlay. Insets show higher magnification images of the MHP (yellow box) and DLHP (blue box). Annotations: arrow = MHP; arrowhead = DLHP; dotted line (in J-L) = interface between the NE and NNE layers of the neural folds. Scale bars: 25 μ m.

mean=299.412. 5 som: Untreated: n=33, mean=154.688; DMSO: n=30, mean=161.747; blebbistatin-treated: n=23, mean=217.472. Two-sided Mann Whitney U test: 5 som - untreated vs DMSO: P=0.332; untreated vs blebbistatin-treated: P=1.30e⁻⁷; DMSO-treated vs blebbistatin-treated: P = 3.50e⁻⁶. 7 som: untreated: n=28, mean=90.444; DMSO-treated: n=27, mean 84.855; blebbistatin-treated: n=24, mean= 169.779. Two-sided Mann Whitney U test: 7 som - untreated vs DMSO: P=0.170; untreated vs blebbistatin-treated: P=2.06e⁻⁹; DMSO-treated vs blebbistatin-treated: P = 1.46e⁻⁹. Annotations: double white arrows = cell length in deep layer; open arrowhead = DLHP; asterisks = rounded neuroectodermal cells; red double arrow = posterior-most telencephalon width. Scale bars: 25 μ m in A and a1; 100 μ m in B.

Reviewers' comments:

Reviewer #1 (Remarks to the Author):

The authors have addressed all point raised by the different referees. Overall, the revised manuscript has improved over the original one; however, there is still very little insight provided into the mechanistic conservation of neural keel/tube formation between zebrafish and other vertebrates undergoing primary neurulation. That said, the phenotypic analysis is at least consistent the main claims of evolutionary conservation made in this study, and thus the manuscript could in principle be published as is.

Reviewer #3 (Remarks to the Author):

Werner and colleagues have reasonably addressed some of the points raised during the first round of review, and –as a result of this- the manuscript has considerably improved. Nevertheless, the study (as stands now) still remains mostly descriptive and in my view, and therefore lacking the necessary mechanistic insights to consolidate major author's conclusions. In this regard, therefore, there are some points that should still be addressed:

a) In an attempt to understand tissue dynamics during anterior neurulation, authors generated time lapse transverse imaging:

a.1) Supp. Movie 1: MHP cells sink inwards as neural folds elevate. Although this movie nicely shows that collective cell movements is indeed required for midline neural plate formation, there is no actually analysis of cell behaviour underlying this process (i.e. tracking, rates of cell convergence, orientation, etc.) neither quantification of time-dependent cell intercalation across layers (not obvious from the present movie/manuscript). This latter point should be clearly addressed to fully support authors interpretation based on previous studies (Ivanovicht et al., 2013). In the same line, the quantification of neural fold elevation in Fig. 8J should properly describe the elevation process (i.e. rates of fold deformation over time) and not as "a distance between the basal side of neural fold cells and the apical side of MHP cells" as this latter is not fixed and change over time.

a.2) Supp. Movie 4: Neural fold cells constrict basally as neural folds elevate. I think the evidence of a "novel cell behaviour -basal constriction of neural fold cells- that contributes to neural fold elevation" is still too premature. In Fig. 4 authors have illustrated few examples selected from 2D imaging data and of course it might not necessarily represent the true scenario. In addition, it does not entirely support the cell rearrangements observed in similar stages (Fig. 9-10). To fully demonstrate this point, I would suggest to authors to use 3D reconstruction analysis over time (i.e. apical surface area vs total cell length) as well as EM. In addition, if basal constriction is somehow required for neural fold, then author should provide functional data to probe this point.

b) The functional data in the paper is still scarce and not conclusive. i) While authors have provided evidence of markers for apical polarity in the anterior neural plate, without functional studies (i.e. gain and loss of function) it is very hard to understand the role of these apical protein markers for the anterior zebrafish neurulation, ii) The PCP phenotype in *vangl2*^{-/-} embryos (provide in Fig. A of the rebuttal letter) is also unclear for the present story. It is well-known that these mutant embryos undergo a series of morphogenetic defects (i.e. delayed converge) and therefore this data might mislead the main cellular mechanisms actually explored in this work, iii) Authors also report an interesting oscillatory surface behaviour of medial neural plate cells, however due to the the lack of

proper functional data I can't see how this cell behaviour may contribute to the story, iv) if DLHPs formation is myosin-independent, then authors should study how apical constriction occurs at these region (i.e. study F-actin).

RESPONSE TO REVIEWERS

We thank Reviewers 1 and 3 for their additional comments on our manuscript entitled "Hingepoints and neural folds reveal conserved features of primary neurulation in the zebrafish forebrain" and provide a complete response to their comments, with a focus on specific requests by the editors at *Communications Biology* to:

1. Improve some of the analyses, as Reviewer 3 suggested
2. Do an analysis of cell behavior and quantification of time-dependent cell intercalation across the layers
3. Quantify neural fold elevation in Figure 8J
4. Tone down some statements/conclusions

We feel that we have now addressed all of the points – with the exception of measurements of rates of individual cell intercalation (see below) - and believe that, with these changes, the manuscript meets the standards for publication in *Communications Biology*.

The blue and orange font corresponds to edits for the first and the second round of editing, respectively.

Response to Reviewer 1

The authors have addressed all point raised by the different referees. Overall, the revised manuscript has improved over the original one; however, there is still very little insight provided into the mechanistic conservation of neural keel/tube formation between zebrafish and other vertebrates undergoing primary neurulation. That said, the phenotypic analysis is at least consistent [with] the main claims of evolutionary conservation made in this study, and thus the manuscript could in principle be published as is.

Response: We believe that the lack of molecular mechanistic insight is offset by the novelty and significance of the findings regarding the conservation of cellular and morphological aspects of neurulation.

Response to Reviewer 3

1. Werner and colleagues have reasonably addressed some of the points raised during the first round of review, and - as a result of this - the manuscript has considerably improved. Nevertheless, the study (as stands now) still remains mostly descriptive and in my view, and therefore lacking the necessary mechanistic insights to consolidate major author's conclusions. In this regard, therefore, [there] are some points that should still be addressed.

Response: With respect to the comment about the descriptive nature of the work, we would like to emphasize that mechanistic insight is not a criterion for publication in *Communications Biology*. The stated aim of this journal is to publish research articles that "represent significant advances bringing new biological insight to a specialized area of research". We feel that we have met this criterion, as our manuscript corrects a long-held view that teleosts have evolved a unique mode of neurulation. By demonstrating the presence of neural folds and hingepoint structures in the anterior neural plate of zebrafish and providing evidence for the functionality of the medial hingepoint (MHP), we have revealed a deeper level of conservation of the cell behaviors underlying neurulation than previously recognized. In the Discussion section of the paper we highlight this conservation while toning down the language to reflect the current limited mechanistic understanding of underlying processes.

2. In an attempt to understand tissue dynamics during anterior neurulation, authors generated time lapse transverse imaging:

a.1) Comment regarding Supplemental Movie 1: MHP cells sink inwards as neural folds elevate. Although this movie nicely shows that collective cell movements is indeed required for midline neural plate formation, there is no actually analysis of cell behaviour underlying this process (i.e. tracking, rates of cell convergence, orientation, etc.) neither quantification of time-dependent cell intercalation across layers (not obvious from the present movie/manuscript). This latter point should be clearly addressed to fully support authors interpretation based on previous studies (Ivanovicht et al., 2013). In the same line, the quantification of neural fold elevation in Fig. 8J should properly describe the elevation process (i.e. rates of fold deformation over time) and not as "a distance between the basal side of neural fold cells and the apical side of MHP cells" as this latter is not fixed and changes over time.

Response: We now provide two additional time-lapse movies that have enabled us to trace the MHP cells from their formation in the superficial layer of the anterior neural plate to their incorporation into the deep layer of the eye field, consistent with the findings of Ivanovitch et al, 2013. However, owing to the lack of single cell resolution in our time-lapse movies - our Zeiss LSM 900 Confocal microscope does not allow sufficient imaging resolution of deep layers in optical transverse views to visualize a clear outline of individual cells - it has not been possible to measure rates of individual cell radial intercalation. While informative, the latter measurement is not as critical as the fate mapping information, which we now provide. The text edits relevant to this new experiment can be found on p. 11, line 215 and read as follows: “The MHP is a transient structure, given that it is no longer observed by 7 som. To fate map this cell population, we photoconverted mKaede in the dorsal and medial region of the ANP (n = 2 embryos, Supplemental Movie 3 and Figure 4G-I). We observed that these cells sink inwards and become incorporated into the deep layer of the eye field”.

Using these new movies and our two original ones, we further provide tissue-level measurements of neural fold convergence, neural fold elevation, neural groove formation, basal neural fold angle and optic vesicle evagination – analyzing how these values change relative to one another over time and rates of change (Figure 8).

Reviewer 3 raised an important point regarding the measurement of neural fold elevation (previously Figure 8J), which should be based on a fixed reference point. We now use the position of the basal surface of the neural fold at time zero (T0 in Figure 8A), prior to the onset of elevation, as a reference point and measure the change in height of the basal surface of the neural fold relative to it.

a.2) Supp. Movie 4: Neural fold cells constrict basally as neural folds elevate. I think the evidence of a “novel cell behaviour -basal constriction of neural fold cells- that contributes to neural fold elevation” is still too premature. In Fig. 4 authors have illustrated few examples selected from 2D imaging data and of course it might not necessarily represent the true scenario. In addition, it does not entirely support the cell rearrangements observed in similar stages (Fig. 9-10). To fully demonstrate this point, I would suggest to authors to use 3D reconstruction analysis over time (i.e. apical surface area vs total cell length) as well as EM. In addition, if basal constriction is somehow required for neural fold, then author should provide functional data to probe this point.

Response: We agree with Reviewer 3 that we currently lack evidence to make a strong conclusion about whether basal cells actively constrict or not. However, the images (Figure 2b1-c1 (red circles) and Figure 4) clearly show that neural fold cells adhere to a focal point on the basal lamina and we have therefore changed the language to reflect this:

p. 10, line 206: *Replaced* “We observed that as the NFs elevate (blue dotted line in Figure 4A-C), NF cells basally constrict (red outline and arrow in Figure 4a1-c1). Interestingly, this cell behavior appears restricted to cells within the *emx3* expression domain (Figure 4D-f1)” with “We observed that as the NFs elevate (blue dotted line in Figure 4A-C), **the basal surface of NF cells narrows** (red outline and arrowhead in Figure 4a1-c1). Interestingly, this **cell shape change** appears restricted to cells within the *emx3* expression domain (Figure 4D-f1).

p. 11, line 212: *Replaced* “[These findings] further identify basal constriction of NF cells as a cell shape change that contributes to NF formation” with “[These findings] ... further reveal **that NFs have a “reverse hinge point” cytoarchitecture, with a narrow basal pole anchored on a focal point and an expanded apical surface.**”

p. 19, line 409: *Replaced* “We instead identify basal constriction as a NF-intrinsic cell behavior that functions as a “reverse hinge point”. Basal constriction may contribute to the early stages of NF elevation or ridging/kinking (implicated in the acquisition of the bilaminar topology of the NFs). Based on scanning EM images of chick embryos, it appears that this cell behavior may also be conserved in amniotes” with “We instead identify **narrowing of the basal surface of NF cells as a cell shape change that occurs concomitantly with elevation of the NFs.** Based on scanning EM images of chick embryos, it appears that **this “reverse hinge point” structure is also observed in amniotes and is linked to neural fold elevation or ridging/kinking**^{16,17}. **It is tempting to speculate that narrowing of the basal surface of NF cells is driven by an active process such as basal constriction.**”

In response to additional comments from Reviewer 3 on NF formation, we would like to raise a few points of clarification: 1) “basally constricted cells” were observed in two separate movies, on both sides of the embryo, thus, we have not selectively represented data; 2) Figures 9 and 10 show the morphology of the anterior neural plate at 5 somites, when the neural folds have already delaminated. In contrast, Figure 4 A-f1 shows neural fold dynamics at 2-3 somites, prior to delamination, when the basal surface of the epithelium undergoes “ridging and kinking”. The apparent discrepancy between these figures therefore merely reflects the different developmental stage of the embryos that were imaged. While the cells retain their “basally constricted” morphology at 5 somites, this cytoarchitecture is best captured with higher resolution mosaic labeling, as shown in Figure 2b1; 3) all of our movies are 3D, as we image across y, x and z dimensions over time, however the type of 3D reconstruction Reviewer 3 has in mind is not readily feasible as the apical surface of neural fold cells is highly protrusive and the neural fold cells approach the midline at an oblique angle 4) Freeze fracture-EM, which would be required to image cells in transverse views, is a very specialized technique that is not routinely and readily performed in zebrafish, as evidenced by the dearth of data published using this technology; 5) providing functional data on basal constriction extends beyond the scope of this paper but is definitely an area we plan to further investigate in the future.

b) The functional data in the paper is still scarce and not conclusive. i) While authors have provided evidence of markers for apical polarity in the anterior neural plate, without functional studies (i.e. gain and loss of function) it is very hard to understand the role of these apical protein markers for the anterior zebrafish neurulation, ii) The PCP phenotype in *vangl2*^{-/-} embryos (provide in Fig. A of the rebuttal letter) is also unclear for the present story. It is well-known that these mutant embryos undergo a series of morphogenetic defects (i.e. delayed converge) and therefore this data might mislead the main cellular mechanisms actually explored in this work, iii) Authors also report an interesting oscillatory surface behavior of medial neural plate cells, however due to the lack of proper functional data I can't see how this cell behaviour may contribute to the story, iv) if DLHPs formation is myosin-independent, then authors should study how apical constriction occurs at these region (i.e. study F-actin).

Response: Reviewer 3 raises excellent suggestions regarding functional analyses, however we feel that these extend well beyond the scope of this already lengthy paper – indeed, each of the functional analyses Reviewer 3 proposes has, in the published literature, been the point of focus of an entire publication. To further elaborate: i) and ii) The challenge in performing loss-of-function studies, as illustrated by the issues we ourselves raised regarding *vangl2* mutants, is that they would necessitate conditional knockout - both temporal and spatial - to get around the pleiotropic effect of this mutation. Gain-of-function studies have not typically been used to characterize genes implicated in MHP formation and would necessitate considerable extended research to complete successfully. These studies therefore represent, in our opinion, worthy future lines of investigation, iii) there is abundant evidence in the literature for how oscillatory behaviors contribute to apical constriction (PMID: 19029882, 24803648, 19563762), including in the frog neural plate, which is in many regards quite similar to that of zebrafish (PMID: 28219946) – we therefore do not feel that it is a stretch to propose that such behaviors could contribute to MHP formation in zebrafish; iv) the mechanism via which the DLHPs constrict is controversial and there is no general consensus across vertebrate models of neurulation (PMID: 26079577 and references within), thus, while of great interest, this question would necessitate extensive additional research.

In summary, our study reveals conservation of mechanisms of neurulation at the morphological and cellular level and provides some, albeit limited insight into molecular mechanisms (apical markers, blebbistatin studies). We feel that the language used in the Discussion section of the paper aligns with these limitations; e.g. page 19, line 394: “These observations suggest that the actomyosin machinery is used across vertebrate to drive cranial neural tube closure and it will be interesting in the future to test whether upstream regulators of apical constriction such as Shroom3¹³ are also conserved.”

UPDATED FIGURES

Figure 4. Dynamics of neural fold formation and MHP intercalation. A-F. Time lapse movie frames of the ANP, from a transverse view. A-C. Still frames of an embryo expressing membrane Kaede (mKaede). D-F. Still frames of a *Tg[emx3:YFP]* embryo expressing membrane RFP (mRFP, pseudo labeled magenta) and YFP (green). Yellow boxes in A, C, D and F identify magnified areas in a1, c1, d1 and f1. G-I: Still frames of an embryo expressing mKaede (green), in which the MHP cells have been photoconverted (magenta) to follow their fate. Abbreviations: MHP: medial hinge point; OV: optic vesicles; hyp: hypothalamus. Annotations: blue dotted line: outlines the basal side of the neural folds; red and white lines identify individual neural fold cells; arrowhead: indicate narrowing surface in neural folds cells; arrows: show direction of intercalation of MHP cells into the eye field; dotted line: separation of the superficial and deep layers. Scale bars: 50 μm in A, D and G; 25 μm in a1 and d1.

Figure 8. Dynamics of anterior neurulation. A. Still frames of an embryo expressing mKaede, showing how the measurements in graphs B-L were acquired, at a discrete time point. Neural groove was measured as the angle formed by the dorsal most tissue; the optic vesicle angle was measured as the angle formed by the outline of the optic vesicle as it evaginates; the neural fold basal angle was measured as the angle formed by the basal surface of NF cells; the neural fold elevation was measured as the difference between the (elevated) position of the basal surface of NFs relative to their initial position at time zero (T0); the distance between the neural folds is the measure of the distance between the basal surface of NF cells. B-G: Graphs illustrating the dynamics of neural groove formation (left Y axis, blue line in B and E), optic vesicle angle (left Y axis, blue line in C and F) and neural fold basal angle (left Y axis, blue line in D and G) as compared to neural fold elevation (right Y axis, black line in B-D) and distance between the neural folds (right Y axis, black line in E-G) over time (X axis). H-L: Measurements indicated in (A) were converted into rates: (measurement frame 2 – measurement frame 1) / time_step. Solid lines represent a fitted linear model for rate measurements of 4 embryos with the standard error as the shaded area. Scale bar in A: 50 μm.

REVIEWERS' COMMENTS:

Reviewer #3 (Remarks to the Author):

The authors have addressed some of the points raised by the previous round of reviews. Using mostly existing movies authors have re-analyzed their data in the light of previous suggestions although some of the important analysis (i.e. time-dependent cell intercalation across the neural plate) is still missing. In addition, while authors have toned down some of their main statements and conclusions, the authors should be less dogmatic about their claim on the evolution of neurulation. Authors mention that their work “corrects a long-held view that teleosts have evolved a unique mode of neurulation. By demonstrating the presence of neural folds and hinge point structures in the anterior neural plate of zebrafish and providing evidence for the functionality of the medial hinge point (MHP), we have revealed a deeper level of conservation of the cell behaviors underlying neurulation than previously recognized”. It would be fair to say some aspects of teleost neurulation are reminiscent of neurulation in other animals and this suggests the teleost mechanism is not totally unique. But it should also be made clear that many aspects of the morphogenesis of the teleost neural plate are substantially different (indeed unique) in comparison to other animals. Since teleost neurulation is likely to be a derived characteristic from an ancestor that used conventional neurulation mechanics, it is perhaps not surprising if it retains some aspects of the ancestral condition. This should be discussed in a more balanced way.

My major concern is still the lack of mechanistic insight of this work (as stated in my previous report...see section: b) The functional data in the paper is still scarce and not conclusive.). Taking together, the study is interesting but still remains mostly descriptive and still lacks the necessary mechanistic insights to consolidate author's major conclusions.

1. We would first like to point out that the text quoted by Reviewer 3 - “ ... corrects a long-held view that teleosts have evolved a unique mode of neurulation. By demonstrating the presence of neural folds and hinge point structures in the anterior neural plate of zebrafish and providing evidence for the functionality of the medial hinge point (MHP), we have revealed a deeper level of conservation of the cell behaviors underlying neurulation than previously recognized” - was in direct response to his/her comment and was never featured in our manuscript.

2. Reviewer 3 also wrote that “it would be fair to say some aspects of teleost neurulation are reminiscent of neurulation in other animals and this suggests the teleost mechanism is not totally unique. But it should also be made clear that many aspects of the morphogenesis of the teleost neural plate are substantially different (indeed unique) in comparison to other animals. Since teleost neurulation is likely to be a derived characteristic from an ancestor that used conventional neurulation mechanics, it is perhaps not surprising if it retains some aspects of the ancestral condition. This should be discussed in a more balanced way”.

In response to these comments, we have made the following in-text changes:

Title:

The title now reads “**Hallmarks of primary neurulation are conserved in the zebrafish forebrain**” to reflect the suggestion made in the checklist. We chose to leave out the comment about the hindbrain because primary neurulation also occurs in this brain region (albeit in absence of hinge points and neural folds).

Abstract – p. 2 line 26:

We replaced “Our findings reveal a deeper level of conservation of neurulation than previously recognized and establish the zebrafish as a model to understand human neural tube development” with “These results reveal similarities between neurulation in teleosts and other vertebrates and hence the suitability of zebrafish to understand human neurulation”. Please note that this response takes into account suggestions in the Final Revisions Instructions and the Abstract word limit.

Discussion – p.18, line 371:

We added the underlined sentence: “We report here on mechanisms of forebrain morphogenesis in the zebrafish embryo and reveal that cell behaviors typically attributed to primary neurulation are observed in this brain region, namely, the use of hinge points and NFs. These findings lend further credence to the view that primary neurulation is the ancestral condition” (Handrigan (2003).

Discussion – p.20, line 436:

Our concluding sentence was already conservative and has been slightly reworded (to correct grammar). It now reads as follows: “In summary, we reveal that zebrafish forebrain neurulation presents multiple similarities with primary neurulation but also some unique features. These findings contribute to our understanding of the evolution of neurulation and highlight the relevance of zebrafish to understand human neural tube development.”

Methods:

“**Zebrafish strains/ husbandry**” section, p22 l443:

We have added the following: Embryos from developmental stages of 2-10 somites were used (approximately 10-14 hours post fertilization). The sex of the embryos used is unknown.

“Whole-mount *in situ* hybridization and imaging” section, p24 l498:

We have added a few sentences to describe the protocol followed.

Embryos were fixed in 4% PFA, dehydrated in methanol and rehydrated stepwise in methanol/PBS then 100% PBT (1x PBS 0.1% Tween 20). Embryos were incubated in hybridization buffer for 1 hour followed by hybridization at 70°C overnight. Following washes in hybridization buffer/SSC, embryos were incubated with preabsorbed alkaline-phosphatase coupled anti-digoxigenin antibody from Roche (Sigma aldrich, SKU 11093274910), at 1:5000 final dilution overnight at 4°C. Detection was performed using an alkaline phosphatase substrate solution (NBT, Sigma Aldrich SKU 11383213001; BCIP, Sigma Aldrich, SKU 11383221001). Reaction was stopped by washing in PBT.

“Confocal microscopy” section, p25 l521:

We have added the underlined to describe the protocol:

Dorsal view time-lapse microscopy: Embryos were imaged using a Leica confocal microscope (Leica SP5 TCS 4D) at 15 sec/frame capturing <.5µm of tissue. All fluorescently labeled sections were imaged using a Leica confocal microscope (Leica SP5 TCS 4D). Dechorionated live embryos were embedded in <1 mm holes bored in 1.2% low melting agarose in E3 medium, solidified on a glass-bottom dish with size 1.5 coverslips, as previously described⁶¹. Embryos were oriented such that the anterior side was pressed against the glass.

Transverse view time-lapse microscopy: Embryos were imaged using a Zeiss confocal microscope (Zeiss 900 LSM with Airyscan 2), at 20X. Dechorionated live embryos were embedded in 1X low melt agarose in E3 medium, in a glass bottom dish with size 1.5 coverslips. Embryos were oriented such that the ventral side was pressed against the glass.